# From Extraction to Deduction: Resolving Functional Misalignment in RAG via a Collaborative Critic-Reasoner Framework

Yufei Chen[1]   Yao Wang[1]   Haibin Zhang[1]   Hualin Zhou[2]   Tao Gu[2]

## Abstract

Retrieval-augmented generation (RAG) systems suffer from a fundamental functional misalignment where retrievers optimize for semantic relevance, often recalling documents with high background utility but factually erroneous answer spans that generators blindly adopt as cognitive shortcuts. To resolve this, we propose the collaborative CRITIC-REASONER framework that shifts robustness control from coarse-grained filtering to fine-grained cognitive decoupling. We disentangle the generation process into two serialized roles by deploying a CRITIC to perform surgical evidence purification through identifying and masking misleading entities while preserving supportive background context, followed by a REASONER that switches from rote extraction to deductive reasoning based on the residual evidence. We operationalize this framework via a two-stage alignment strategy combining supervised fine-tuning (SFT) with path-aware direct preference optimization (DPO) to enforce strict behavioral synergy. Experimental results on adversarial benchmarks such as ConFiQA demonstrate that our method significantly outperforms baselines, achieving a 25.99% accuracy gain in conflicting scenarios and effectively resolving the trust bias dilemma in real-world RAG.

## 1. Introduction

RAG (Lewis et al., 2020) has emerged as a pivotal paradigm for enhancing large language models (LLMs) by incorporating external contextual knowledge, effectively mitigating hallucinations and knowledge obsolescence (Kandpal et al., 2023; Shuster et al., 2021; Gao et al., 2023). However,

real-world retrieval is often imperfect (Su et al., 2024; Weijia et al., 2023). Compounded by the inevitable noise in knowledge bases, retrieved contexts frequently suffer from factual errors or irrelevant distractions (Yoran et al., 2023). Under such conditions, RAG may not merely fail to yield improvements but can actively mislead the generation process, causing it to deviate from its internal knowledge and correct reasoning trajectories. This failure mode, where noisy evidence hijacks cognitive processes, poses a critical challenge to the robustness of RAG systems.

Prior work has sought to improve RAG robustness through two pathways: retrieval and generation. On the retrieval side, methods such as TrustRAG (Zhou et al., 2025) and CrAM (Deng et al., 2025) attempt to filter or down-weight low-quality documents using clustering or confidence estimation. On the generation side, SelfRAG (Asai et al., 2024) introduces self-reflection tokens to control output quality, while frameworks such as Astute RAG (Wang et al., 2025), CK-PLUG (Bi et al., 2025b), and Knowledgeable-R1 (Lin et al., 2025) prioritize leveraging parametric knowledge to resist contextual interference. While effective in certain settings, these approaches typically operate at the document level and thus force a coarse-grained binary decision: either accept the retrieved document wholly or discard it entirely.

This coarse granularity gives rise to a fundamental dilemma: (1) Resource waste. Discarding a partially correct document wastes valuable background evidence (e.g., definitions, causal relationships) that is crucial for inference. (2) Fallback risk. Conversely, rejecting external documents often forces a fallback to parametric knowledge. In time-sensitive scenarios (e.g., dynamic political events), this fallback inevitably reverts to outdated training data, contradicting RAG's core objective of leveraging external knowledge.

We attribute this dilemma to a structural problem that we term **functional misalignment.** Our empirical analysis reveals a systematic disconnect between the two primary components of RAG: the retriever and the generator. The retriever is optimized for semantic relevance, aiming to maximize similarity between queries and documents, but it is largely insensitive to the factual correctness of specific entities. The generator, by contrast, tends to rely on explicit answer spans as cognitive shortcuts, effectively treating re-

[1]School of Cyber Engineering, Xidian University, Xi'an, China
[2]School of Computing, Macquarie University, Sydney, Australia. Correspondence to: Yao Wang <wangyao@xidian.edu.cn>.

*Proceedings of the 43rd International Conference on Machine Learning*, Seoul, South Korea. PMLR 306, 2026. Copyright 2026 by the author(s).

trieval as an extraction task. Consequently, when a retrieved document contains accurate background but an incorrect answer the retriever ranks it highly while the generator is misled. Existing RAG architectures lack the granularity needed to resolve these conflicting signals.

To bridge this functional gap, we propose a collaborative CRITIC-REASONER framework. Unlike prior methods that apply coarse-grained down-weighting to entire documents, our approach enables fine-grained control over distinct context components. We decompose the generation process into two serialized roles dynamically enacted by a single LLM: The CRITIC performs fine-grained context scrutiny and employs a [MASK] mechanism to precisely isolate potentially misleading answer spans while preserving supportive background context. The REASONER is then trained to generate answers from this partially masked context. By blocking the direct copy-paste shortcut via masking, the REASONER is compelled to shift from mechanical span extraction to deductive reasoning, aggregating distributed background clues to infer the correct answer.

To ensure the behavioral synergy of these components, we introduce a two-stage alignment strategy. Building upon supervised fine-tuning (SFT), we incorporate direct preference optimization (DPO) to operationalize the CRITIC's editing signals as reasoning directives. This alignment enables the REASONER to dynamically modulate its strategy by shifting between direct citation, evidence-based deduction, or parametric fallback depending on evidence availability. Our main contributions are as follows:

- We identify the functional misalignment in RAG systems and propose the collaborative CRITIC-REASONER framework. By employing a filter-then-mask strategy, we decouple error suppression from context retention, enabling a paradigm shift from rote extraction to logical reasoning grounded in residual evidence.

- We develop a two-stage fine-tuning framework that builds upon SFT for foundational capabilities and incorporates path-aware DPO to explicitly model the CRITIC's editing signals as reasoning constraints for the REASONER, ensuring deep behavioral synergy.

- Extensive evaluations across six fine-grained context types and nine datasets demonstrate that our method achieves superior performance. Specifically, in the most challenging adversarial scenario (incorrect answer, relevant background), our method delivers a 25.99% accuracy improvement compared to strong baselines.

## 2. Related Work

RAG mitigates hallucination and knowledge obsolescence by grounding LLMs in external knowledge bases. However, the retrieval of factually erroneous or low-quality documents remains a persistent challenge. Existing defense mechanisms for RAG are mainly categorized into the following two paradigms:

**Retrieval-centric filtering.** This stream focuses on purifying the retrieval pool before generation. CAR (Weller et al., 2024) employs query augmentation to expand the candidate set, subsequently aggregating multi-source documents via voting to dilute erroneous information. TrustRAG (Zhou et al., 2025) introduces a conflict resolution mechanism, clustering retrieved documents to filter out potentially low-quality or contradictory evidence based on consensus. Similarly, CrAM (Deng et al., 2025) estimates document-level confidence scores to modulate attention weights, thereby suppressing the impact of unreliable documents during the encoding phase.

**Generation-centric resilience.** This paradigm aims to enhance the generator's ability to discern and resist noise. SelfRAG (Asai et al., 2024) trains a unified LLM to generate self-reflection tokens, enabling dynamic critique of retrieval quality and adaptive generation. InstructRAG (Wei et al., 2024) uses instruction tuning to improve the model's noise immunity. Focusing on knowledge conflicts, training-free frameworks like AstuteRAG (Wang et al., 2025) and CK-PLUG (Bi et al., 2025b) elicit the LLM's internal parametric knowledge to override external errors. More recently, Knowledgeable-R1 (Lin et al., 2025) employs reinforcement learning to bias the model towards parametric memory when encountering severe contextual interference.

**Limitations.** Existing research is primarily constrained to document-level quality modeling. By treating retrieved contexts as monolithic entities, they force a coarse-grained decision that overlooks the critical distinction between semantic relevance and factual correctness. This oversight leads to the binary dilemma identified in Sec. 1, where discarding such hybrid documents wastes valuable reasoning support, while accepting them exposes the model to misleading shortcuts. Unlike these approaches, our work introduces fine-grained evidence manipulation, enabling the precise isolation of errors without sacrificing the surrounding context.

## 3. Preliminary

### 3.1. Problem Formulation

In the standard RAG framework, let $q$ denote the user query. The retriever $\mathcal{R}$ selects a Top-$k$ document set $\mathcal{D} = \{d_1, \ldots, d_k\}$ from an external knowledge base that maximizes semantic relevance to $q$. The generator $\mathcal{G}$ then conditions on both $q$ and $\mathcal{D}$ to generate the final response $y$ by maximizing the probability $P(y|q, \mathcal{D})$. Crucially, we posit that each retrieved document $d_i \in \mathcal{D}$ is not a monolithic block of text, but a composite entity consisting of two orthogonal components: *(1) Answer-bearing content*

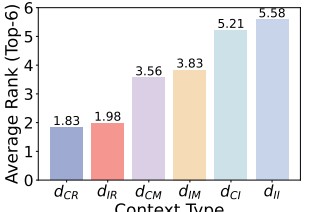 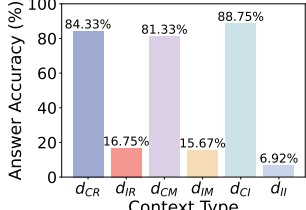

*(a)* Top-6 retrieval rate    *(b)* Answer accuracy

*Figure 1.* Results from the retrieval and generation stages.

*Table 1.* Document quality classification by answer and context.

| Component | $d_{CR}$ | $d_{IR}$ | $d_{CM}$ | $d_{IM}$ | $d_{CI}$ | $d_{II}$ |
|---|---|---|---|---|---|---|
| $c_{ans}$ | Correct | Incorrect | Correct | Incorrect | Correct | Incorrect |
| $c_{bg}$ | Relevant | Relevant | Mixed | Mixed | Irrelevant | Irrelevant |

$(c_{ans})$: The specific text span or fact that directly answers $q$. *(2) Background context ($c_{bg}$):* The supporting narrative or surrounding text that provides context but implies no direct answer. Despite this internal structure, standard RAG assumes a uniform utility for $d_i$. We argue that this assumption leads to a **functional misalignment**, as $\mathcal{R}$ and $\mathcal{G}$ exhibit conflicting sensitivities to these components. To characterize this discrepancy, we propose the following hypotheses:

**H1 (Retrieval bias):** The retriever $\mathcal{R}$ prioritizes the semantic relevance and volume of $c_{bg}$, rendering it insensitive to the factual correctness of $c_{ans}$.

**H2 (Generator reliance):** The generator $\mathcal{G}$ relies critically on the factual correctness of $c_{ans}$ for accurate generation, while treating $c_{bg}$ primarily as auxiliary support.

This disconnect, where $\mathcal{R}$ fetches documents based on background signals while $\mathcal{G}$ demands precise answer spans, constitutes the core robustness challenge we aim to address.

### 3.2. Evaluation Setup

To systematically validate hypotheses H1 and H2, we adopt the conceptual decomposition defined in Sec. 3.1 and conduct a controlled evaluation (data construction and implementation details are provided in Appendix C). We categorize retrieval contexts into 6 fine-grained quality types based on the orthogonal dimensions of answer correctness ($c_{ans}$) and background relevance ($c_{bg}$), as detailed in Table 1.

We construct diagnostic instances for these quality types and analyze them across two stages: *(1) Retrieval stage (testing H1):* We perform vector retrieval on the 6 context types using query $q$ and examine the Top-6 ranking results. This allows us to determine whether the retriever is driven primarily by the relevance of $c_{bg}$ or the correctness of $c_{ans}$. *(2) Generation stage (testing H2):* We feed each context type separately into the generator to produce response $y$ and evaluate answer accuracy. This evaluates the degree of the generator's dependence on $c_{ans}$, as well as its ability to utilize $c_{bg}$ for reasoning.

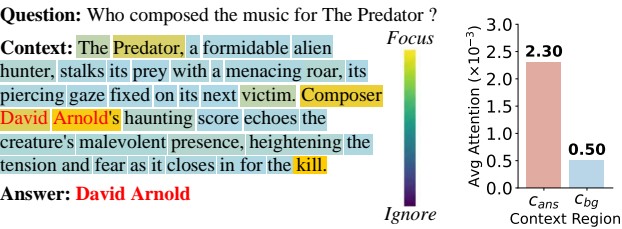

*(a)* Context attention distribution heatmap    *(b)* Attention

*Figure 2.* Analysis of attention distribution over $c_{ans}$ and $c_{bg}$.

### 3.3. Result Analysis

**Retrieval stage findings (support H1).** Experimental results reveal a decisive bias in the retriever's ranking behavior. As shown in Fig. 1a, factual correctness has a negligible impact on ranking. The factually correct $d_{CR}$ has an average ranking of 1.83, with no significant difference from the factually incorrect $d_{IR}$ at 1.98. In contrast, background relevance is the dominant factor. When the background becomes irrelevant ($d_{CI}$), the average rank drops sharply to 5.21. This confirms that the retriever is driven by the semantic relevance of $c_{bg}$ and remains insensitive to the factual variations in $c_{ans}$, supporting H1.

**Generation stage findings (support H2).** The generator exhibits the opposite sensitivity. As illustrated in Fig. 1b, performance is primarily determined by the factual correctness of $c_{ans}$. Under identical background relevance, the correct context $d_{CR}$ yields an accuracy of 84.33%, whereas the incorrect context $d_{IR}$ plummets to 16.75%, with a massive gap of 67.58%. Furthermore, when both components are unreliable ($d_{II}$), accuracy hits a floor of 6.92%. This indicates that coherent background information $c_{bg}$ alone cannot compensate for factual errors in the answer span, strongly supporting H2.

**Mechanistic evidence: attention distribution.** To elucidate the mechanism underlying H2, we analyze the model's attention distribution. Visualizing the character-level heatmap (Fig. 2a) reveals that the model's focus is predominantly fixated on answer-bearing entities (e.g., "David Arnold"), even when they are factually incorrect. To verify this at scale, we conducted a quantitative analysis across 500 sampled contexts using LLaMA3.1-8B-Instruct. We computed the average attention weights from the final decoder layer assigned to $c_{ans}$ versus $c_{bg}$. The results (Fig. 2b) demonstrate that the average attention on $c_{ans}$ is approximately 4.6 times higher than on $c_{bg}$. This disparity mechanistically confirms H2, i.e., the generation process is driven by specific answer spans, rendering the model vulnerable to specific factual errors despite valid background support.

**Conclusion.** In summary, the controlled experiments reveal a critical disconnect: the retriever is driven by background relevance ($c_{bg}$), whereas the generator relies heavily on an-

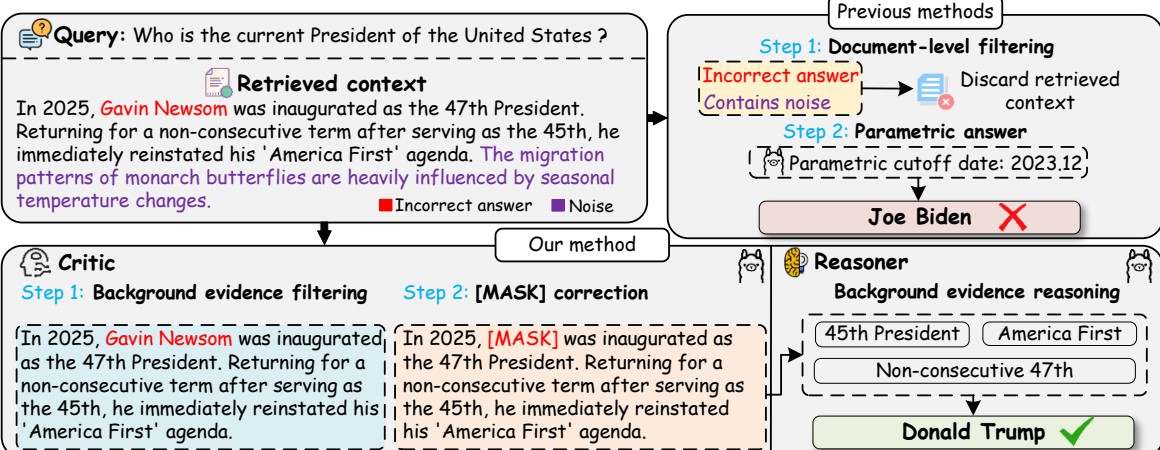

*Figure 3.* **Overview of the collaborative framework.** Instead of discarding imperfect documents entirely, our approach performs surgical evidence purification. The CRITIC masks misleading spans to preserve valid background, enabling the REASONER to deduce correct answers from residual evidence, thereby avoiding hallucinations from outdated parametric memory.

swer correctness ($c_{ans}$). This functional misalignment poses a dilemma for standard RAG systems. Existing document-level filtering methods typically discard contexts like $d_{\text{IR}}$ due to their incorrect answers. We argue that such coarse-grained rejection is suboptimal: it wastes high-quality background evidence that could clearly support reasoning. To address this, we move beyond document-level robustness and develop a mechanism that can selectively utilize valid background information while suppressing erroneous answer spans, motivating the cooperative alignment framework proposed in Sec. 4.

## 4. Our Method

### 4.1. Overview and Task Definition

To bridge the functional gap identified in Sec. 3, we propose that a robust generator model should possess three specific capabilities to handle unreliable retrieval. *(1) Background information filtering (addressing H1):* Since retrievers may recall contexts containing noise, the model should selectively filter and aggregate only the background evidence relevant to the query to maximize the signal-to-noise ratio (SNR). *(2) Incorrect answer suppression (counteracting H2):* Addressing the generator's tendency to blindly trust answer spans, the model should precisely identify and isolate factually incorrect answer fragments to avoid being misled by erroneous shortcuts. *(3) Background information reasoning (resolving resource waste):* Instead of directly discarding documents containing incorrect answers (which would waste valid background evidence), the model should attempt to deduce the correct answer through reasoning by leveraging the remaining valid background clues when explicit answer spans are unreliable.

To this end, we formalize these requirements as a multi-objective alignment problem. Within our framework, a

single LLM plays two complementary roles under different prompt instructions, namely CRITIC and REASONER, to achieve collaborative robustness. Our objective is to optimize the model to simultaneously satisfy two goals: *(1) Answer correctness:* The final generated answer must match the ground truth; *(2) Context utilization robustness:* The generation process must faithfully adhere to the quality of the retrieved evidence, filtering noise (as CRITIC) and reasoning based on valid clues to provide answers (as REASONER). This transforms the RAG task from a simple generation problem into a dual-alignment task, ensuring the model remains robust even when retrieval quality fluctuates.

### 4.2. Cooperative CRITIC-REASONER Architecture

As illustrated in Fig. 3, our approach operationalizes the capabilities defined in Sec. 4.1 by decoupling the generation process into two serialized roles, CRITIC ($\kappa$) and REASONER ($\rho$), orchestrated by a single LLM. This framework implements a coherent workflow of evidence purification followed by context-aware inference:

**The CRITIC ($\kappa$): Filter-then-mask strategy.** It serves as the gatekeeper of evidence quality. Given a query $q$ and a retrieved document $d$, $\kappa$ transforms the raw context into a refined evidence state $\tilde{d}$ via a two-stage protocol:

(1) *Background filtering:* It first filters the background context $c_{\text{bg}}$, discarding segments irrelevant to $q$ to maximize the information SNR.

(2) *Masking correction:* Crucially, identifying potentially erroneous answer spans $c_{ans}$ and replacing them with the special token [MASK]. We strategically adopt masking rather than deletion. Unlike deletion, which fragments the text, [MASK] preserves the syntactic structure and discourse coherence of the original sentence. This structural integrity is

vital, as it explicitly signals the locus of unreliability to the downstream REASONER while maintaining the surrounding context necessary for logical inference.

**The REASONER ($\rho$): Robust contextual inference.** It generates the final answer $\hat{y} = \rho_\theta(q, \tilde{d})$ conditioned on the edited evidence. By acting on the output of $\kappa$, $\rho$ is forced to shift its operational mode from rote extraction to deductive reasoning. Specifically, when the explicit answer shortcut is obstructed by [MASK], the model cannot rely on copying specific entities. Instead, it is compelled to aggregate distributed background clues ($c_{bg}$) surrounding the [MASK] token to derive the correct answer. This mechanism effectively physicalizes the suppression of generator reliance (H2), enhancing both the robustness and interpretability of the generation process.

## 4.3. Joint Supervised Fine-Tuning (SFT)

To enable the acquisition of dual capabilities (CRITIC and REASONER) within a single generative model, we construct a multi-task instruction-tuning dataset (refer to Appendix B.1) and optimize the shared parameters $\theta$ via joint training. This procedure ensures the model stably switches roles based on different system prompts (for the prompts of the CRITIC and REASONER, see Appendix D), bridging the gap between evidence quality and reasoning requirements.

**SFT for the CRITIC: evidence refinement.** $\kappa$'s objective is to transform the raw retrieval $d$ into a refined evidence state $\tilde{d}$. To establish the filter-then-mask logic, we synthesize training samples focusing on three dimensions:

- Background filtering (signal enhancement). To enable the extraction of salient evidence from high-noise environments, we supervise $\kappa$ to identify and retain supporting facts while eliminating irrelevant context. This explicitly maximizes the information SNR of the context $c_{bg}$.

- Masking correction (error isolation). We construct parallel corpora containing factual truths and counterfactual distractors. For contexts with factual errors ($c_{ans}$), the model is trained to precisely localize the error and replace it with [MASK]. Conversely, for factually correct contexts, the model learns an identity mapping, preserving the original text to avoid over-correction.

- Extreme rejection (hallucination prevention). To enhance robustness, we introduce negative samples where both $c_{ans}$ and $c_{bg}$ are invalid. In such scenarios, the CRITIC is forced to output an empty state $\varnothing$, preventing downstream hallucination from completely unreliable sources.

It is optimized via maximum likelihood estimation:

$$\mathcal{L}_{\text{SFT}}^\kappa(\theta) = -\mathbb{E}_{(x_\kappa, \tilde{d}) \sim \mathcal{D}_{\text{SFT}}^\kappa} \big[ \log \pi_\theta^\kappa(\tilde{d} \mid x_\kappa) \big],$$

where inputs $x_\kappa = (q, d)$ and targets $\tilde{d}$ are drawn from the mixed CRITIC dataset $\mathcal{D}_{\text{SFT}}^\kappa$.

**SFT for the REASONER: robust inference.** $\rho$ learns to derive the final answer $y$ based on the CRITIC-edited evidence $\tilde{d}$. We classify the supervision signals into two categories to foster adaptive reasoning:

- Context-aware reasoning (learning from partial evidence). We utilize a teacher model (e.g., GPT-4o) to synthesize chain-of-thought (CoT) reasoning traces derived specifically from masked contexts. This mechanism compels the model to bypass the masked shortcut and instead aggregate dispersed background clues in $\tilde{d}$ to deduce the answer, enforcing the utilization of $c_{bg}$.

- Parametric fallback (adaptive knowledge switching). For cases where $\tilde{d}$ provides insufficient evidence (e.g., insufficient information to uniquely identify the target entity), we align the model's output with its internal parametric knowledge. We set the target to a fallback response $y^{fb}$ generated by the model itself without retrieval. This endows the REASONER with the capability to fall back smoothly when external evidence is deemed unreliable.

The REASONER's objective is formulated as:

$$\mathcal{L}_{\text{SFT}}^\rho(\theta) = -\mathbb{E}_{(x_\rho, \hat{y}) \sim \mathcal{D}_{\text{SFT}}^\rho} \big[ \log \pi_\theta^\rho(\hat{y} \mid x_\rho) \big],$$

where $x_\rho = (q, \tilde{d})$ and $\hat{y}$ is either the ground truth or the fallback answer.

**Multi-task joint training.** We construct a unified training set $\mathcal{D}_{SFT} = \mathcal{D}^\kappa \cup \mathcal{D}^\rho$ by mixing samples from both tasks. The shared parameters $\theta$ are then optimized via a weighted joint loss:

$$\mathcal{L}_{\text{SFT}}(\theta) = \mathcal{L}_{\text{SFT}}^\kappa(\theta) + \alpha \, \mathcal{L}_{\text{SFT}}^\rho(\theta),$$

where $\alpha$ is a hyperparameter balancing the gradient contributions. This joint formulation enables the model to simultaneously acquire the discriminative capability of the CRITIC and the robust generative capability of the REASONER, while ensuring implementation efficiency by deploying a single model for the entire pipeline.

## 4.4. Preference-based CRITIC-REASONER Alignment

Although SFT equips the single model with the fundamental capabilities of both the CRITIC $\kappa$ and the REASONER $\rho$, a critical challenge of behavioral inconsistency remains. SFT maximizes the likelihood of static tokens but does not explicitly penalize the REASONER for ignoring the CRITIC's structural signals (e.g., generating an answer span that was just masked). To address this, we employ direct preference optimization (DPO) to align the shared parameters $\theta$, enforcing a strict functional dependency where the REASONER's strategy dynamically adapts to the evidence availability defined by the CRITIC (i.e., the edited document $\tilde{d}$).

**Path-aware preference construction.** We treat $\kappa$'s output $\tilde{d}$ as a structured control signal and define three path selection rules to determine the preferred reasoning trajectory:

- Rule 1: Direct answering. If $\tilde{d}$ does not contain `[MASK]` (typically $\tilde{d} \approx d$), implying the retrieved evidence is reliable, the REASONER should adopt a direct citation mode, extracting the answer from the verified context.
- Rule 2: Background reasoning. If `[MASK]` $\in \tilde{d}$, indicating specific answer spans are unreliable, the REASONER should reduce reliance on the masked tokens and instead aggregate the filtered background evidence $c_{bg}$ to deduce the answer.
- Rule 3: Parametric fallback. If $\tilde{d} = \varnothing$, indicating the document is entirely rejected, the REASONER switches to parametric answering (fallback) to avoid hallucination under non-retrieval conditions.

Guided by this collaboration mechanism where $\tilde{d}$ acts as a switch among the three answering modes, we construct path-specific preference pairs for DPO. Specifically, for each training sample $(q, \tilde{d})$, we select the ground truth reasoning path (which inherently satisfies both factual correctness and the corresponding path rule) as the preferred output $y^+$. Conversely, we sample a response that violates the rule as the non-preferred output $y^-$, constructing the dataset $\mathcal{D}_{\text{DPO}}$.

**Reward instantiation.** We formally characterize the reward function with two complementary objectives:

$$R(q, \tilde{d}, y) = R_{\text{ans}}(y, y^*) + \beta\, R_{\text{use}}(q, \tilde{d}, y),$$

where $y$ denotes the generated response and $y^*$ represents the ground truth. Specifically, $R_{\text{ans}}(y, y^*)$ measures answer correctness (e.g., via exact match) against the ground truth, and $R_{\text{use}}$ is a binary consistency check that evaluates whether the generation path adheres to the active path selection rule. The hyperparameter $\beta$ controls the trade-off, encouraging the model to effectively leverage evidence when available while strictly avoiding fabrication when $\tilde{d} = \varnothing$.

**DPO optimization.** We optimize the policy $\pi_\theta$ to maximize the margin between the rule-following path $y^+$ and the rule-violating path $y^-$. The objective is minimized as:

$$\mathcal{L}_{\text{DPO}}(\theta) = -\mathbb{E}_{(x, y^+, y^-) \sim \mathcal{D}_{\text{DPO}}} \left[ \log \sigma \left( \beta_{\text{dpo}}(\Delta_\theta - \Delta_{\text{ref}}) \right) \right],$$

$$\text{where} \begin{cases} \Delta_\theta = \log \pi_\theta(y^+ \mid x) - \log \pi_\theta(y^- \mid x) \\ \Delta_{\text{ref}} = \log \pi_{\text{ref}}(y^+ \mid x) - \log \pi_{\text{ref}}(y^- \mid x). \end{cases}$$

Here, $x = (q, \tilde{d})$, $\sigma(\cdot)$ is the sigmoid function, and $\beta_{\text{dpo}}$ is the KL penalty coefficient. Without explicitly modeling a reward function, this objective implicitly aligns the model's preference to favor safe reasoning paths derived from the CRITIC's judgment.

**Theoretical analysis.** We provide theoretical motivation for our DPO by analyzing the optimal solution of the underlying KL-regularized objective. In particular, for input states containing unreliable signals, there exists a sufficient threshold on the usage-weight such that the optimal policy assigns higher probability to the rule-following response than to the rule-violating one, provided the preference pair is separable by the usage-rule reward. This result characterizes how tuning the usage-weight encourages robust reasoning paths; see Appendix A for the formal statement and derivation.

## 5. Evaluation

We empirically validate our proposed framework's ability to mitigate functional misalignment between the retriever and generator. Specifically, we aim to answer three key research questions: (1) Does the CRITIC effectively filter noise and isolate erroneous answer spans? (2) Can the REASONER leverage residual background evidence to deduce correct answers when explicit shortcuts are blocked? (3) Does the system smoothly fall back to parametric knowledge when external evidence is entirely unreliable?

### 5.1. Experimental Setup

**Dataset construction.** To systematically evaluate model robustness, we categorize retrieval contexts into 6 fine-grained quality types based on two orthogonal dimensions: answer span correctness ($c_{ans}$) and background relevance ($c_{bg}$), as defined in Table 1. To instantiate these scenarios, we leverage four representative datasets. ConFiQA (Bi et al., 2025a) serves as the backbone for counterfactual retrieval, providing both QA (single-hop) and MC (multi-hop) subsets to simulate factual conflicts. Furthermore, to replicate high-noise environments where reasoning is required, we incorporate HotpotQA (Yang et al., 2018), 2WikiMHQA (Ho et al., 2020), and MuSiQue (Trivedi et al., 2022), which contain correct answers embedded within substantial irrelevant background noise. In addition, we introduce Natural Questions (NQ) (Kwiatkowski et al., 2019) as a real-world benchmark beyond the training distribution, in order to verify the generalization ability of our method on unseen open-domain QA scenarios. The specific construction is as follows:

(1) $d_{\text{CR}}$: We utilize the *orig_context* from the ConFiQA-QA subset to construct **CR-QA**, and the original Wikipedia passages from NQ to construct **CR-NQ**, both providing correct answers within relevant contexts. (2) $d_{\text{IR}}$: We use the *cf_context* from the ConFiQA-QA subset to form **IR-QA**, and construct **IR-NQ** by replacing the correct entities in CR-NQ with incorrect ones, thereby introducing factual conflicts while preserving background relevance. (3) $d_{\text{CM}}$: We employ HotpotQA, 2WikiMHQA, and MuSiQue to represent scenarios where correct answers are embedded within substantial irrelevant noise. (4) $d_{\text{IM}}$: Based on IR-QA, we further inject irrelevant noise to construct **IM-QA**, creating an adversarial setting with both incorrect answers and a mixed background. (5) $d_{\text{CI}}$: We synthesize **CI-QA** by embedding the correct answer span from the QA subset into an irrelevant context sampled from unrelated queries. (6)

*Table 2.* Answer accuracy across six fine-grained scenarios: $d_{CR}$, $d_{IR}$, $d_{CM}$, $d_{IM}$, $d_{CI}$, $d_{II}$. Best results are shown in **bold**, second-best results are underlined. The "Improve" row shows the gain relative to the second-best result.

| Method | $d_{\mathbf{CR}}$ | | $d_{\mathbf{IR}}$ | | $d_{\mathbf{CM}}$ | | | $d_{\mathbf{IM}}$ | $d_{\mathbf{CI}}$ | $d_{\mathbf{II}}$ |
|---|---|---|---|---|---|---|---|---|---|---|
| | CR-QA | CR-NQ | IR-QA | IR-NQ | HotpotQA | 2Wiki | MuSiQue | IM-QA | CI-QA | II-MC |
| *Llama3.1-8B-Instruct* | | | | | | | | | | |
| RAG prompting | 78.32% | 81.10% | 38.00% | 21.30% | 38.83% | 48.33% | 40.17% | 34.42% | 52.33% | 16.08% |
| InstructRAG (Wei et al., 2024) | 31.83% | 81.40% | 40.17% | 20.45% | 48.75% | 57.42% | 50.42% | 22.08% | 44.67% | 23.67% |
| CK-PLUG (Bi et al., 2025b) | 41.08% | 30.75% | 35.14% | 33.65% | 20.17% | 32.25% | 8.17% | 44.58% | 44.14% | 23.25% |
| AstuteRAG (Wang et al., 2025) | 77.97% | 72.00% | 43.08% | 25.75% | 48.17% | 57.58% | 45.25% | 40.42% | 50.10% | 34.42% |
| GRPO w/ RAG (Lin et al., 2025) | **82.62%** | 77.10% | 39.26% | 33.75% | 34.84% | 41.22% | 16.59% | 52.75% | 49.66% | 35.69% |
| Knowledgeable-R1 (Lin et al., 2025) | 80.03% | 78.60% | 44.59% | 31.70% | 37.06% | 45.37% | 14.69% | 50.33% | 51.33% | 41.12% |
| **Our method** | 80.14% | **81.60%** | **70.58%** | **63.55%** | **59.83%** | **67.25%** | **55.92%** | **67.58%** | **56.62%** | **74.00%** |
| **Improve** | -2.48% | +0.20% | +25.99% | +29.80% | +11.08% | +9.67% | +5.50% | +14.83% | +4.29% | +32.88% |
| *Qwen2.5-7B-Instruct* | | | | | | | | | | |
| RAG prompting | 74.58% | 73.70% | 33.17% | 26.75% | 41.00% | 42.00% | 38.92% | 32.67% | 51.67% | 26.42% |
| InstructRAG (Wei et al., 2024) | 78.11% | 74.45% | 25.92% | 27.65% | 51.00% | 57.47% | 48.42% | 33.42% | 51.75% | 24.50% |
| CK-PLUG (Bi et al., 2025b) | 41.92% | 39.70% | 40.58% | 37.80% | 22.42% | 36.08% | 9.67% | 34.83% | 41.25% | 34.83% |
| AstuteRAG (Wang et al., 2025) | 77.25% | 67.20% | 17.33% | 11.85% | 23.83% | 33.08% | 26.25% | 14.67% | 41.92% | 13.08% |
| GRPO w/ RAG (Lin et al., 2025) | 80.03% | 72.15% | 26.01% | 24.70% | 27.93% | 33.95% | 11.79% | 33.25% | 52.25% | 19.74% |
| Knowledgeable-R1 (Lin et al., 2025) | **80.90%** | 73.15% | 29.40% | 25.85% | 31.45% | 37.52% | 12.04% | 32.67% | 50.75% | 37.34% |
| **Our method** | 78.33% | **74.80%** | **63.50%** | **57.35%** | **59.00%** | **60.67%** | **53.75%** | **53.00%** | **54.92%** | **76.33%** |
| **Improve** | -2.57% | +0.35% | +22.92% | +19.55% | +8.00% | +3.20% | +5.33% | +18.17% | +2.67% | +38.99% |

$d_{II}$: Using the *cf_context* from the ConFiQA MC subset, where both the answer is incorrect and the background is irrelevant, we construct **II-MC**.

**Models and baselines.** We evaluate our method using Llama3.1-8B-Instruct (Dubey et al., 2024) and Qwen2.5-7B/3B-Instruct (Yang et al., 2024) as backbones. We benchmark against 5 state-of-the-art robust RAG frameworks: Knowledgeable-R1 (Lin et al., 2025), GRPO with RAG (Lin et al., 2025), Astute-RAG (Wang et al., 2025), CK-PLUG (Bi et al., 2025b), and InstructRAG (Wei et al., 2024), See Appendix D for baseline prompts. These baselines primarily focus on document-level quality assessment. To ensure rigorous comparability, all models are trained and evaluated under identical datasets and configurations, with details provided in Appendix B.

**Evaluation metrics.** We assess answer accuracy using a combination of exact match, token-level F1 score (with a threshold of $\geq 0.7$), and GPT-4o assisted judgment (prompt in Appendix D). A prediction is considered correct if it satisfies any of these three criteria. Additionally, we propose the background utilization rate (BUR) to quantify the model's reasoning capability. This metric measures the proportion of answers derived from background evidence through reasoning, serving as a proxy for the REASONER's responsiveness to masking and reflecting the success rate of the [MASK] mechanism.

**5.2. Main Results**

5.2.1. ANSWER ACCURACY ANALYSIS

Table 2 presents a comparative analysis of answer accuracy across 6 fine-grained context qualities. We analyze the

results through four critical capability dimensions to validate the robustness of our framework.

**Retention of performance upper bound** ($d_{\mathbf{CR}}$). A robust system must not degrade performance on high-quality retrieval while seeking robustness. In the ideal scenario ($d_{CR}$), our method maintains parity with the strongest baselines, achieving 80.14% accuracy on CR-QA and 81.60% on CR-NQ (Llama3.1-8B). This confirms that the additional CRITIC-REASONER overhead does not compromise fundamental generation capabilities when evidence is accurate.

**Resilience to misinformation** ($d_{\mathbf{IR}}$, $d_{\mathbf{IM}}$). This dimension highlights our core contribution: mitigating the generator's blind reliance on retrieved errors. In $d_{IR}$, where retrieved documents contain factually incorrect answer spans, standard RAG collapses to 38.00% on IR-QA and merely 21.30% on the out-of-distribution IR-NQ. In contrast, our method achieves 70.58% and 63.55% respectively, surpassing the strongest baselines by 25.99% and 29.80%. This confirms that the robustness transfers to unseen real-world distributions rather than overfitting to training data. Under the more adversarial $d_{IM}$, where incorrect answers are further obscured by noise, we sustain 67.58% accuracy, outperforming GRPO w/ RAG by 14.83%. These results validate the CRITIC's efficacy in masking misleading shortcuts, forcing the REASONER to derive answers via background reasoning rather than copying erroneous spans.

**Noise filtration in low-SNR environments** ($d_{\mathbf{CM}}$). The model's ability to filter information is rigorously tested when correct answers are buried in substantial noise. Our approach demonstrates significant advantages here, achieving improvements of 11.08%, 9.67%, and 5.50% over best-

*Table 3.* Background Utilization Rate (BUR) performance across six fine-grained scenarios.

| Model | Metrics | $d_{\mathbf{CR}}$ | | $d_{\mathbf{IR}}$ | | $d_{\mathbf{CM}}$ | | | $d_{\mathbf{IM}}$ | $d_{\mathbf{CI}}$ | $d_{\mathbf{II}}$ |
|---|---|---|---|---|---|---|---|---|---|---|---|
| | | CR-QA | CR-NQ | IR-QA | IR-NQ | HotpotQA | 2Wiki | MuSiQue | IM-QA | CI-QA | II-MC |
| *Llama3.1-8B-Instruct* | BUR | 2.33% | 13.20% | **96.42%** | **78.15%** | 0.92% | 0.75% | 0.25% | **91.17%** | 1.67% | 0.42% |
| *Qwen2.5-7B-Instruct* | BUR | 1.25% | 28.60% | **94.25%** | **84.80%** | 0.58% | 1.11% | 0.75% | **81.92%** | 0.75% | 0.12% |

*Table 4.* Ablation study for different components of model and SFT data mixture. w/ stands for with, w/o stands for without.

| Model | $d_{\mathbf{CR}}$ | | $d_{\mathbf{IR}}$ | | $d_{\mathbf{CM}}$ | | | $d_{\mathbf{IM}}$ | $d_{\mathbf{CI}}$ | $d_{\mathbf{II}}$ |
|---|---|---|---|---|---|---|---|---|---|---|
| | CR-QA | CR-NQ | IR-QA | IR-NQ | HotpotQA | 2Wiki | MuSiQue | IM-QA | CI-QA | II-MC |
| ***Ablation Study for Different Components of Model*** | | | | | | | | | | |
| Llama3.1-8B-Instruct | 79.14% | 70.00% | 47.62% | 30.50% | 41.17% | 48.68% | 30.75% | 40.17% | 51.33% | 29.33% |
| Llama3.1-8B-Instruct w/ SFT | 57.17% | 71.05% | 52.92% | 22.00% | 39.17% | 53.48% | 51.00% | 34.17% | 52.75% | 43.00% |
| Llama3.1-8B-Instruct w/ DPO | 67.75% | 60.80% | 55.58% | 47.65% | 41.92% | 57.67% | 36.83% | 52.67% | 54.42% | 31.75% |
| Llama3.1-8B-Instruct w/ SFT, DPO | 80.14% | 81.60% | 70.58% | 63.55% | 59.83% | 67.25% | 55.92% | 67.58% | 56.62% | 74.00% |
| ***Ablation Study for SFT Data Mixture*** | | | | | | | | | | |
| Our method w/o Reasoning | 59.33% | 60.10% | 59.90% | 50.25% | 44.58% | 57.33% | 49.67% | 52.67% | 34.50% | 30.67% |
| Our method w/o [MASK] | 50.50% | 32.85% | 51.17% | 31.05% | 29.17% | 13.75% | 14.67% | 51.08% | 49.92% | 68.25% |
| Our method | 80.14% | 81.60% | 70.58% | 63.55% | 59.83% | 67.25% | 55.92% | 67.58% | 56.62% | 74.00% |

performing baselines on HotpotQA, 2Wiki, and MuSiQue, respectively. This performance gap indicates that the CRITIC module effectively functions as an evidence purifier, stripping away irrelevant noise to maximize the SNR for the downstream REASONER.

**Safety against hallucination ($d_{\mathbf{CI}}$, $d_{\mathbf{II}}$).** In the hallucination hotspot where retrieval is completely invalid (incorrect answer and irrelevant background), standard RAG accuracy drops to 16.08%. In contrast, our method achieves an accuracy of 74.00% on II-MC, exceeding the best baseline by 32.88%. This validates the extreme rejection mechanism: when the CRITIC detects a complete absence of evidence ($\tilde{d} \to \varnothing$), it triggers a fallback to parametric knowledge, ensuring factual safety when retrieval fails completely. Furthermore, when facing low-quality evidence (where the context contains the correct answer, but the surrounding background is completely irrelevant to the query), our method is able to effectively filter and enable the parametric fallback mechanism. In the $d_{\mathbf{CI}}$ scenario, we achieve an accuracy of 56.62%, representing an improvement of 4.29% over the baseline RAG prompting.

### 5.2.2. BACKGROUND UTILIZATION RATE

To verify whether our method achieves the intended collaborative mechanism, where the CRITIC isolates erroneous signals and the REASONER performs evidence-based reasoning, we analyze the BUR across six scenarios. The results are reported in Table 3.

**Error correction ($d_{\mathbf{IR}}$, $d_{\mathbf{IM}}$).** The most critical validation comes from scenarios where answer spans are incorrect but the background remains relevant. On Llama3.1-8B-Instruct,

the BUR reaches 96.42% on IR-QA, 78.15% on the out-of-distribution IR-NQ, and 91.17% on IM-QA. A similar pattern is observed on Qwen2.5-7B-Instruct, with BUR of 94.25%, 84.80%, and 81.92% respectively. These consistently high utilization rates, including on the unseen IR-NQ, confirm that the CRITIC successfully masks erroneous fragments, forcing the REASONER to bypass incorrect shortcuts and derive answers strictly from background evidence.

**Correct extraction ($d_{\mathbf{CR}}$, $d_{\mathbf{CM}}$).** In scenarios where the answer spans themselves are correct, the BUR drops significantly (e.g., only 0.25% on MuSiQue). This indicates that the model does not blindly enforce reasoning but instead adopts a direct answer mode, prioritizing the extraction of explicit answer spans.

**Safe rejection ($d_{\mathbf{CI}}$, $d_{\mathbf{II}}$).** When background information is irrelevant or completely invalid, BUR remains negligible (e.g., only 0.42% on II-MC). This confirms that the model does not hallucinate from noise, but instead ensures generation safety through parametric fallback.

### 5.2.3. ABLATION EXPERIMENTS

**Different components.** To evaluate each component's impact on performance, we conducted ablation experiments using Llama3.1-8B-Instruct. Results appear in Table 4. Using SFT or DPO alone provides foundational capabilities but leads to significant performance degradation. When both SFT and DPO are combined, the model's contextual robustness improves substantially, demonstrating that these components play complementary roles.

**SFT data mixture.** Table 4 shows the performance of our method under different SFT data compositions. Removing

*Table 5.* Comparison with query-only parametric answering.

| Method | $d_{\mathbf{CR}}$ | | $d_{\mathbf{IR}}$ | | $d_{\mathbf{CM}}$ | | | $d_{\mathbf{IM}}$ | $d_{\mathbf{CI}}$ | $d_{\mathbf{II}}$ |
|---|---|---|---|---|---|---|---|---|---|---|
| | CR-QA | CR-NQ | IR-QA | IR-NQ | HotpotQA | 2Wiki | MuSiQue | IM-QA | CI-QA | II-MC |
| *Llama3.1-8B-Instruct* | | | | | | | | | | |
| Query-only prompting | 54.17% | 47.60% | 54.17% | 47.60% | 30.25% | 40.42% | 10.58% | 54.17% | 54.17% | 34.00% |
| Our method | 80.14% | 81.60% | 70.58% | 63.55% | 59.83% | 67.25% | 55.92% | 67.58% | 56.62% | 74.00% |
| *Qwen2.5-7B-Instruct* | | | | | | | | | | |
| Query-only prompting | 44.67% | 39.00% | 44.67% | 39.00% | 23.33% | 27.17% | 7.58% | 44.67% | 64.67% | 30.17% |
| Our method | 78.33% | 74.80% | 63.50% | 57.35% | 59.00% | 60.67% | 53.75% | 53.00% | 54.92% | 76.33% |

*Table 6.* Ablation study for different model scales

| Method | $d_{\mathbf{CR}}$ | | $d_{\mathbf{IR}}$ | | $d_{\mathbf{CM}}$ | | | $d_{\mathbf{IM}}$ | $d_{\mathbf{CI}}$ | $d_{\mathbf{II}}$ |
|---|---|---|---|---|---|---|---|---|---|---|
| | CR-QA | CR-NQ | IR-QA | IR-NQ | HotpotQA | 2Wiki | MuSiQue | IM-QA | CI-QA | II-MC |
| *Qwen2.5-3B-Instruct* | | | | | | | | | | |
| RAG prompting | 76.33% | 64.35% | 21.92% | 18.50% | 36.38% | 43.57% | 37.58% | 20.92% | 45.50% | 21.83% |
| Our method | 76.72% | 60.85% | 61.50% | 43.20% | 50.75% | 54.00% | 46.67% | 47.75% | 47.50% | 76.75% |
| Improve | +0.39% | -3.50% | +39.58% | +24.70% | +14.37% | +10.43% | +9.09% | +26.83% | +2.00% | +54.92% |
| *Qwen2.5-7B-Instruct* | | | | | | | | | | |
| RAG prompting | 74.58% | 73.30% | 33.17% | 26.75% | 41.00% | 42.00% | 38.92% | 32.67% | 51.67% | 26.42% |
| Our method | 78.33% | 74.80% | 63.50% | 57.35% | 59.00% | 60.67% | 53.75% | 53.00% | 54.92% | 76.33% |
| Improve | +3.75% | +1.50% | +30.33% | +30.60% | +18.00% | +18.67% | +14.83% | +20.33% | +3.25% | +49.91% |

either the reasoning dataset or the masking dataset degrades performance. The impact is especially severe on 2Wiki, where removing the `[MASK]` dataset causes accuracy to drop by 53.5%. These results demonstrate that both datasets are essential for the model's performance.

**Model parameters.** Table 5 compares our method's performance against answering based solely on the model's internal parameters. Our method consistently outperforms the parametric baseline across all datasets. On the dataset $d_{\mathrm{IR}}$, accuracy improves by 16.41%. This gain demonstrates that our method breaks through the performance ceiling of existing approaches that fall back to parametric answering when retrieval quality is poor, achieving effective retrieval augmentation even in noisy environments.

**Model scale.** To investigate how model scale affects our framework's performance, we extended experiments to the smaller Qwen2.5-3B-Instruct model, with results in Table 6. Our method remains highly effective even on lightweight models. In the most challenging $d_{\mathrm{IR}}$ scenario, accuracy improved by 39.58% compared to the standard RAG baseline, validating our architecture's effectiveness across different parameter scales.

We discuss the limitations of our method, generation efficiency (time and tokens), and error propagation in Appendix E, as well as conduct case studies in Appendix F.

## 6. Conclusion

We resolved the functional misalignment in RAG systems by introducing the collaborative CRITIC-REASONER framework. By shifting from coarse-grained filtering to surgical evidence purification, our method precisely isolates misleading shortcuts while retaining high-utility background. Empowered by a two-stage SFT and DPO alignment, the model transitions from rote extraction to deductive reasoning. Empirical results confirm a 25.99% accuracy gain in adversarial scenarios, validating that fine-grained cognitive decoupling, rather than binary document rejection, is the critical path toward robust RAG systems.

## Acknowledgments

We are grateful to the Area Chair and the anonymous reviewers for their rigorous evaluation and insightful feedback. Their expertise and detailed comments were instrumental in improving the technical depth and presentation of this manuscript. This work was supported in part by the Natural Science Basic Research Plan in Shaanxi Province of China under Program No. 2025JC-JCQN-089, in part by the Fundamental Research Funds for the Central Universities under No. YJSJ26003, and in part by the Xidian University Special Research Fund for Interdisciplinary Exploration under Grant TZJHF202501.

## Impact Statement

This paper presents a framework aimed at enhancing the robustness and reliability of RAG systems. By mitigating the risk of functional misalignment and reducing hallucinations caused by misleading context, our work contributes to the deployment of safer AI systems in high-stakes domains such as medical consultation, legal advisory, and financial analysis. We believe that transitioning from extractive mimicry to deductive reasoning is a crucial step toward trustworthy AI. Researchers and practitioners should weigh the additional computational and deployment costs of this multi-step framework against its safety benefits when deploying such systems.

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

# A. Proof of Theorem

In this section, we provide the formal statement and proof of the robustness guarantee discussed in Section 4.4.

**Theorem A.1**(Robustness Guarantee). Let $x = (q, \tilde{d})$ be an input state with unreliable signals (e.g., $\texttt{[MASK]} \in \tilde{d}$), and consider two candidate responses: $y_{\text{safe}}$ (rule-following) and $y_{\text{risky}}$ (rule-violating). Suppose the safe response achieves a strictly higher context-usage reward, i.e., $\Delta R_{\text{use}} > 0$.

Then, there exists a threshold $\beta_{\text{th}}$ such that for any robustness weight $\beta > \beta_{\text{th}}$, the optimal policy $\pi^*$ satisfies:

$$\pi^*(y_{\text{safe}} \mid x) > \pi^*(y_{\text{risky}} \mid x). \tag{1}$$

A sufficient choice for this threshold is derived as:

$$\beta_{\text{th}} = \frac{-\Delta R_{\text{ans}} - \beta_{\text{dpo}} \ln \left( \frac{\pi_{\text{ref}}(y_{\text{safe}} \mid x)}{\pi_{\text{ref}}(y_{\text{risky}} \mid x)} \right)}{\Delta R_{\text{use}}}. \tag{2}$$

We now provide the formal derivation for the sufficient condition established in the theorem above, demonstrating that the optimal policy $\pi^*$ strictly prioritizes the safe reasoning path over the risky one.

*Proof.* Following the KL-regularized reward maximization framework underlying DPO (Rafailov et al., 2023), the optimal policy admits the closed-form solution:

$$\pi^*(y \mid x) = \frac{1}{Z(x)} \pi_{\text{ref}}(y \mid x) \exp\left( \frac{1}{\beta_{\text{dpo}}} R(x, y) \right), \tag{3}$$

where $x = (q, \tilde{d})$ denotes the input state, $\pi_{\text{ref}}$ is the reference policy, $Z(x)$ is the partition function, and $\beta_{\text{dpo}} > 0$ is the DPO KL coefficient.

We seek a condition ensuring $\pi^*(y_{\text{safe}} \mid x) > \pi^*(y_{\text{risky}} \mid x)$. Substituting (3) into the inequality and cancelling the partition function $Z(x)$ yields:

$$\pi_{\text{ref}}(y_{\text{safe}} \mid x) \exp\left( \frac{R(x, y_{\text{safe}})}{\beta_{\text{dpo}}} \right) > \pi_{\text{ref}}(y_{\text{risky}} \mid x) \exp\left( \frac{R(x, y_{\text{risky}})}{\beta_{\text{dpo}}} \right). \tag{4}$$

Rearranging terms and taking the natural logarithm on both sides gives:

$$\beta_{\text{dpo}} \ln \frac{\pi_{\text{ref}}(y_{\text{safe}} \mid x)}{\pi_{\text{ref}}(y_{\text{risky}} \mid x)} + \left( R(x, y_{\text{safe}}) - R(x, y_{\text{risky}}) \right) > 0. \tag{5}$$

Recall the composite reward definition $R(x, y) = R_{\text{ans}}(y, y^*) + \beta R_{\text{use}}(x, y)$. We define the reward differences as:

$$\Delta R_{\text{ans}} \triangleq R_{\text{ans}}(y_{\text{safe}}, y^*) - R_{\text{ans}}(y_{\text{risky}}, y^*), \tag{6}$$

$$\Delta R_{\text{use}} \triangleq R_{\text{use}}(x, y_{\text{safe}}) - R_{\text{use}}(x, y_{\text{risky}}). \tag{7}$$

By assumption, the safe response correctly adheres to the usage rules while the risky one violates them, implying $\Delta R_{\text{use}} > 0$. Substituting these definitions into (5), we obtain:

$$\beta_{\text{dpo}} \ln \frac{\pi_{\text{ref}}(y_{\text{safe}} \mid x)}{\pi_{\text{ref}}(y_{\text{risky}} \mid x)} + \Delta R_{\text{ans}} + \beta \Delta R_{\text{use}} > 0. \tag{8}$$

Since $\Delta R_{\text{use}} > 0$, we can solve the inequality for $\beta$ without reversing the sign:

$$\beta > \underbrace{\frac{-\Delta R_{\text{ans}} - \beta_{\text{dpo}} \ln \frac{\pi_{\text{ref}}(y_{\text{safe}} \mid x)}{\pi_{\text{ref}}(y_{\text{risky}} \mid x)}}{\Delta R_{\text{use}}}}_{\beta_{\text{th}}}. \tag{9}$$

Thus, for any usage weight $\beta > \beta_{\text{th}}$, the inequality holds, implying $\pi^*(y_{\text{safe}} \mid x) > \pi^*(y_{\text{risky}} \mid x)$. $\qquad\square$

# B. Training Dataset and Experimental Setup

## B.1. Training Dataset

The dataset construction is as follows. We construct supervised samples based on multiple public datasets, enabling a single generator model to play the roles of critic and reasoner under different prompt templates, and collaboratively complete robust answering. To ensure fair experimental comparison, our training and test sets both adopt the data configuration from the baseline method Knowledgeable-R1 (Lin et al., 2025).

**Training CRITIC Dataset:**

(1) **Background Evidence Filtering.** Using HotpotQA to construct evidence filtering data, training the model to filter and aggregate background information relevant to the query in contexts containing substantial noise, thereby enhancing robustness against interfering information.

(2) **Masking Correction.** Based on the cf_context from ConFiQA subsets QA where the answer is incorrect but the background is relevant, replacing incorrect answer fragments with [MASK] to construct masking correction supervision data, training the model to identify and isolate unreliable answer cues. Meanwhile, introducing orig_context from ConFiQA subsets QA where the answer is correct and the background is relevant as positive supervision examples to constrain the model to keep the original text intact when the context is fully reliable, avoiding excessive masking.

(3) **Extreme Rejection.** Using cf_context from ConFiQA subset MC where the answer is incorrect and the background is irrelevant to construct extreme negative samples, training the model to output a rejection signal $\varnothing$ when the context is entirely unavailable, thereby reducing the risk of hallucination or forced generation under unreliable evidence.

**Training REASONER Dataset:**

(1) **Background-grounded Reasoning.** Based on the masked context generated from the "mask correction" data, we invoke GPT-4o to generate aligned step-by-step reasoning traces given the background evidence and target answer, constructing reasoning supervision samples to train the model to primarily rely on background evidence to complete reasoning and generate answers when [MASK] is present.

(2) **Parametric Fallback.** For samples where GPT-4o still cannot derive the correct answer given the masked context, we construct "reasoning failure" data: supervising the model to fall back to parametric answering based solely on the query $q$ when evidence is insufficient or reasoning cannot be completed, thereby improving the system's stability and safety in non-inferable scenarios.

## B.2. Experimental Setup

**Training Setup.** SFT Phase, all models use a learning rate of $5 \times 10^{-5}$, batch size of 4, gradient accumulation of 8, and are trained for 3 epochs with a maximum sequence length of 2048 tokens. DPO Phase, the learning rate is set to $5 \times 10^{-6}$, batch size of 2, gradient accumulation of 16, and trained for 1 epoch with the same 2048 tokens setting. All experiments are completed for training and testing on an A800 GPU, and use the same system prompt during evaluation to ensure result comparability.

**Testing Setup.** Experiments test the models on a single-card environment based on the vLLM inference engine, loading models with bfloat16 precision. Generation parameters are set to temperature 0.1, top-p 0.9, with a maximum generation length limited to 512 tokens.

# C. Preliminary Evaluation Setup

**Dataset Construction.** ConFiQA-QA is a counterfactual retrieval benchmark that provides orig_context with correct answers and relevant background, as well as cf_context with incorrect answers but relevant background. Additionally, the dataset provides variants containing only correct answers (orig_answer) and only incorrect answers (cf_answer). Based on ConFiQA-QA, we construct the six fine-grained context types shown in Table 1, with the specific construction methods as follows:

(1) $d_{\text{CR}}$: Uses orig_context from ConFiQA-QA.

(2) $d_{\text{IR}}$: Uses cf_context from ConFiQA-QA.

(3) $d_{CM}$: Constructed by concatenating orig_context from ConFiQA-QA with irrelevant context sampled from other questions.

(4) $d_{IM}$: Constructed by concatenating cf_context from ConFiQA-QA with irrelevant context sampled from other questions.

(5) $d_{CI}$: Constructed by embedding the correct answer to the current target query (orig_answer) into an irrelevant context sampled from disparate, unrelated queries.

(6) $d_{II}$: Constructed by applying the same substitution mechanism, but injecting the incorrect answer to the current target query (cf_answer) into the irrelevant context sampled from unrelated queries.

**Retrieval phase.** We use `bge-large-en` (Chen et al., 2024) to construct the `FAISS` vector database.

**Generation phase.** We use `LLaMA3.1-8B-Instruct` as the generator, with temperature 0.1, and a maximum generation limit of 50 tokens.

## D. Prompt Templates

---

**Prompt for RAG**

You are a helpful assistant, below is a query from a user and some relevant contexts. Answer the question given the information in those contexts. If you cannot find the answer to the question, just say "I don't know".

---

*Figure 4.* RAG prompting.

---

**Prompt for InstructRAG**

Your task is to analyze the provided documents and answer the given question.
Please generate a brief explanation of how the contents of these documents lead to your answer.
If the provided information is not helpful to answer the question, you only need to respond based on your own knowledge, without referring to the documents."

Document [1] (Title: Context): {context}
Based on your knowledge and the provided information, answer the question: {question}

Below are some examples of how to answer the question:
Document [1] (Title: Geography): Paris is the capital and most populous city of France.
Based on your knowledge and the provided information, answer the question: What is the capital of France?
Output: Paris

---

*Figure 5.* Prompt for InstructRAG.

---

**Prompt for Astute RAG**

You are a helpful assistant. After the user asks a question, you first think carefully and then give the answer.

When responding, please keep the following points in mind:
Answer the question using consolidated information from both internal and external documents.
The reasoning process and answer are enclosed within <think> </think> and <answer> </answer> tags, respectively.
Output your final answer directly between the tag <answer> </answer> without any intermediate steps.
If the user gives a multiple choice question, your answer must be a single option A or B or C or D or E

Here is an example:
Question: what is the capital of China?
<think> The provided documents indicate that Beijing is the capital of China. </think>
<answer> BeiJing </answer>
Now, you should answer user's question. After answer user's question, you should stop generate.

---

*Figure 6.* Prompt for Astute RAG.

---

**Prompt for GRPO**

You are a helpful assistant. After the user asks a question, you first think carefully and then give the answer.

When responding, please keep the following points in mind:
The reasoning process and answer are enclosed within <think> </think> and <answer> </answer> tags, respectively.
Output your final answer directly between the tag <answer> </answer> without any intermediate steps.

Here is an example:
user's question: what is the capital of China?
<think> reasoning process here </think>
<answer> BeiJing </answer>
Retrieved information: {retrieved information}
Question: {question}

---

*Figure 7.* Prompt for GRPO with RAG.

---

**Prompt for Knowledgeable-R1**

You are a helpful assistant. After the user asks a question, you first think carefully and then give the answer.

When responding, please keep the following points in mind:
The reasoning process and answer are enclosed within <think> </think> and <answer> </answer> tags, respectively.
Output your final answer directly between the tag <answer> </answer> without any intermediate steps.

Here is an example:
user's question: what is the capital of China?
</think>The capital of China is Beijing. This is a well-known geographical fact. </think>
<answer> BeiJing </answer>

---

*Figure 8.* Prompt for Knowledgeable-R1.

---

**Prompt for Reasoner.**

You are an expert reasoning assistant. Answer the Question by evaluating the reliability of the provided Context and selecting the appropriate strategy.

Follow the decision rules below and output JSON only.
Decision rules:
1) If Context is NULL: treat it as irrelevant and answer using internal knowledge.
2) If Context contains `[MASK]`: the answer span is hidden; infer the answer from remaining background clues.
3) Otherwise: treat Context as clean evidence; answer by extracting or directly supported reasoning from the text.

Output format (JSON only):
Analysis: {
Context status: null | masked | clean,
Background information span: Context text if available; otherwise None ,
Inference process: Step-by-step reasoning, e.g., 1) identify clues; 2) apply logic; 3) conclude.
},
Status: Success | Insufficient_info,
Correct answer: Final answer string

---

*Figure 10.* Prompt for REASONER.

---

**Prompt for Critic**

You are an advanced fact-checking assistant. Your task involves three steps:

1. Analyze the full Context and extract the specific background information relevant to the Question, assigned to relevant_background. If the Context is irrelevant or fully incorrect regarding the Question, set relevant_background to null.

2. Verify whether the answer provided in relevant_background is factually correct or incorrect or hallucinated.

3. Generate a JSON response with:
Relevant background: Contextual background information related to the question, or null if no relevant information exists.

Factuality status: 'correct' if accurate, 'incorrect' if it contains a wrong answer, or null if 'relevant_background' is null.

Detected error: The incorrect entity string extracted from 'relevant_background' if status is 'incorrect', else null.

Final context: The 'relevant_background' string with the 'detected_error' replaced by '[MASK]'. If status is 'correct', output 'relevant_background' as is. If status is null, output null.

---

*Figure 9.* Prompt for CRITIC.

---

**Prompt for GPT-4o assisted judgment**

Task: Determine if [Model Answer] is correct based on [Ground Truth Options].
  Input:
  - Question: {question}
  - Ground Truth Options: {ground_truth_list_str}
  - Model Answer: {pred_answer}

  Logic:
  If the Model Answer matches the meaning of ANY option in Ground Truth, return true.

---

*Figure 11.* Prompt for GPT-4o assisted judgment.

## E. Limitation and Discussion

### E.1. Limitation

**Evaluation Scenarios.** Our evaluation primarily focuses on complex Question Answering (QA) tasks. This scope is driven by two considerations. First, existing baselines for RAG context robustness predominantly utilize QA datasets; adhering to this protocol ensures a direct and rigorous comparative analysis. Second, and crucial to our experimental design, QA tasks provide unambiguous factual ground truth. This precision enables us to manipulate context quality with high granularity—specifically allowing us to isolate answer spans from background evidence—to quantitatively measure the model's robustness against factual errors. While effective for validating our core hypotheses, the applicability of our framework to broader scenarios, such as open-ended generation, multi-turn dialogue, or real-time tool use, remains a subject for future exploration.

### E.2. Discussion

**Generation time and tokens.** Since our framework employs a serialized dual-role architecture (CRITIC-REASONER) to explicitly generate reasoning traces, it inevitably incurs computational overhead compared to standard generation. To quantify this, we measured the average generation time and tokens on 50 randomly sampled instances across different context scenarios, benchmarking against the reasoning-specialized model DeepSeek-R1-Distill-Qwen-7B (DeepSeek-AI, 2025) (results in Table 7).

We observe that our method exhibits adaptive computational expenditure, reflected in both inference latency and token generation volume. As shown in Table 7, the generation is highly efficient in the double-irrelevant scenario ($d_{II}$), consuming

*Table 7.* Generation time and tokens across different datasets

| Method | Metric | $d_{\mathbf{CR}}$ | | $d_{\mathbf{IR}}$ | | $d_{\mathbf{CM}}$ | | | $d_{\mathbf{IM}}$ | $d_{\mathbf{CI}}$ | $d_{\mathbf{II}}$ | Avg |
|---|---|---|---|---|---|---|---|---|---|---|---|---|
| | | CR-QA | CR-NQ | IR-QA | IR-NQ | HotpotQA | 2Wiki | MuSiQue | IM-QA | CI-QA | II-MC | |
| Our method | Time (s) | 3.81 | 3.01 | 5.18 | 3.25 | 4.42 | 4.38 | 4.61 | 5.47 | 2.98 | 1.11 | **3.82** |
| | Tokens | 221 | 213 | 323 | 233 | 301 | 298 | 312 | 343 | 198 | 69 | **251.11** |
| DeepSeek-R1 | Time (s) | 8.46 | 6.03 | 5.49 | 6.27 | 10.25 | 5.52 | 10.44 | 6.74 | 5.59 | 5.97 | **7.08** |
| | Tokens | 555 | 472 | 418 | 492 | 858 | 407 | 732 | 392 | 419 | 442 | **518.7** |

*Table 8.* Robustness analysis of the REASONER under different masking conditions. w/ stands for with.

| Model | CR-QA w/ over-`[MASK]` | CR-QA | IR-QA |
|---|---|---|---|
| Llama3.1-8B-Instruct | 69.50% | 80.14% | 70.58% |
| Qwen2.5-7B-Instruct | 63.25% | 78.33% | 63.50% |

only 1.11s and 69 tokens on average, as invalid inputs are swiftly handled via parametric fallback without unnecessary elaboration. Conversely, in complex deductive scenarios like $d_{\mathrm{IM}}$, the model invests more resources (5.47s, 343 tokens) to construct detailed reasoning chains, ensuring interpretability.

Crucially, our method demonstrates superior token economy compared to the baseline. While DeepSeek-R1-Distill-Qwen-7B averages 518.7 tokens and 7.08s per query, our approach achieves a significantly lower average of 251.11 tokens, resulting in a latency reduction of 3.82s. This indicates that our serialized reasoning architecture is more concise and targeted, avoiding the reasoning bloat often observed in pure reasoning models while maintaining robust performance.

**Resilience to error propagation.** Although the serialized architecture inherently carries the risk of error propagation, our framework can still significantly mitigate this issue through a dual fault-tolerance mechanism. Specifically, when the CRITIC erroneously masks the explicit answer string (false positive) but retains the supporting context, the REASONER is capable of reconstructing the answer through logical reasoning over the remaining background. If the background information is insufficient to support reasoning, the REASONER activates its parametric fallback mechanism to retrieve the answer from parametric knowledge. This hierarchical defense guarantees that the system maintains reasoning-level performance stability, preventing significant degradation even when upstream masking is flawed.

To evaluate the performance of our mechanism under extreme over-masking scenarios, we conducted a test on the CR-QA dataset with correct answers and relevant background: we artificially replaced all correct answer spans with `[MASK]`, retaining only contextual cues, and then tested the REASONER's reasoning accuracy. The experimental results are shown in Table 8.

Experimental results indicate that under the over `[MASK]` scenario, the success rate decreases to 69.50%. This decline is an expected consequence of the cognitive shift from direct answer extraction to deductive reasoning based on background context. However, attributed to our dual resilience mechanism, the framework effectively prevents catastrophic failure. Notably, performance stabilizes at a robust reasoning level, remaining statistically comparable to the IR-QA benchmark (70.58%), where the system autonomously masks incorrect answers. This alignment confirms that our method maintains consistent reasoning capabilities even when explicit cues are entirely removed.

**Resilience to CRITIC false negatives.** A natural concern with the serialized architecture is whether false negatives from the CRITIC (i.e., failing to mask erroneous answer spans) would cause cascading errors in the REASONER. To quantify this risk, we conducted controlled experiments on IR-QA. Specifically, we selected samples where the CRITIC successfully applied `[MASK]`, then progressively restored a proportion of them back to the original erroneous context to simulate increasing false negative rates. The results are reported in Table 9.

*Table 9.* Impact of CRITIC false negative rate on answer accuracy (IR-QA, Llama3.1-8B-Instruct).

| False Negative Rate | 0% | 20% | 40% | 60% | 80% | 100% |
|---|---|---|---|---|---|---|
| Accuracy | 70.17% | 67.08% | 64.17% | 61.75% | 59.25% | 56.58% |

Two key observations emerge. First, each 20% increase in false negative rate leads to only approximately 2.5–3% accuracy degradation, exhibiting a smooth, linear decline with no inflection point or catastrophic collapse. This graceful degradation indicates that the system does not rely on a minimum CRITIC precision threshold to maintain robustness. Second, even at a 100% false negative rate (where no masking is performed and the REASONER receives entirely unedited erroneous contexts), accuracy remains at 56.58%, which still substantially exceeds the standard RAG baseline (38.00% on IR-QA in Table 2). This confirms that the DPO-aligned REASONER has internalized a degree of error resistance beyond what the CRITIC provides.

## F. Case Studies

Figures 12 through 17 present case studies of our method across six distinct scenarios of fine-grained context quality. The specific analysis is as follows:

(1) $d_{\text{CR}}$: In scenarios where the context is accurate and relevant, the CRITIC effectively preserves the correct contextual information, assisting the REASONER in directly utilizing the correct answer within it to complete the question-answering task.

(2) $d_{\text{IR}}$: Facing a context containing an incorrect answer (United States of America) but relevant background, the CRITIC precisely identified and performed a Mask operation on it; the REASONER subsequently successfully utilized the remaining background clues to deduce the correct answer (United Kingdom).

(3) $d_{\text{CM}}$: In mixed contexts containing noise, the CRITIC successfully filtered out irrelevant noise, retaining only the segments relevant to the user query. Accordingly, the REASONER extracted effective information to produce the correct answer.

(4) $d_{\text{IM}}$: For contexts containing both noise and an incorrect answer, the CRITIC not only filtered out the noise but also located and Masked the incorrect entity (American football). Based on the purified background information, the REASONER derived the correct conclusion (Association football) through reasoning.

(5) $d_{\text{CI}}$: When confronting low-quality contexts that contain the correct string but are completely irrelevant in background, the CRITIC judged them as invalid and output null. Upon receiving the null signal, the REASONER activated the fallback mechanism, accurately answering the question using internal parametric knowledge.

(6) $d_{\text{II}}$: Similarly, for contexts with an incorrect answer and irrelevant background, the CRITIC decisively filtered out the entire low-quality text and output null. The REASONER subsequently generated the correct answer using internal parametric knowledge.

| A sample from d$_{CR}$ (correct answer, relevant background) |
| --- |

**Question:** What country is R.S.C. Anderlecht located in?
**Ground Truth Answer:** Belgium.
**Retrieved context:** The club was founded in 1908 and has since become one of the most successful teams in the country, winning numerous domestic and international titles. Anderlecht has a fierce rivalry with fellow Belgian club Club Brugge and has produced many talented players over the years, including international stars like Franck Ribery and Axel Witsel.

| Our method |
| --- |

**Critic:** relevant_background: The club was founded in 1908 and has since become one of the most successful teams in the country, winning numerous domestic and international titles. Anderlecht has a fierce rivalry with fellow Belgian club Club Brugge and has produced many talented players over the years, including international stars like Franck Ribery and Axel Witsel. \n factuality_status: correct \n final_context: The club was founded in 1908 and has since become one of the most successful teams in the country, winning numerous domestic and international titles. Anderlecht has a fierce rivalry with fellow Belgian club Club Brugge and has produced many talented players over the years, including international stars like Franck Ribery and Axel Witsel.

**Reasoner:** 1. [Identify Clue] The context is clean and explicitly contains the answer. 2. [Apply Logic] The phrase "fellow Belgian club" implies that R.S.C. Anderlecht belongs to the same country as Club Brugge, which is explicitly identified as Belgian. 3. [Conclusion] Therefore, the answer is Belgium.

**Final answer:** Belgium.

*Figure 12.* A case study from $d_{CR}$ context type.

| A sample from d$_{IR}$ (incorrect answer, relevant background) |
| --- |

**Question:** What is the country of origin of Our Girl?
**Ground Truth Answer:** United Kingdom.
**Retrieved context:** Our Girl is a American television drama series that premiered on E4 in 2013. Created by Howard Overman, the show follows the life of Molly Dawes, a young woman from a small town in the United States of America who joins the American Army. The series explores themes of friendship, love, and identity as Molly navigates her way through military training and deployment to Afghanistan. With a strong focus on female characters and their experiences, Our Girl has gained a loyal following and critical acclaim for its portrayal of women in the military.

| Our method |
| --- |

**Critic:** Our Girl is a American television drama series that premiered on E4 in 2013. Created by Howard Overman, the show follows the life of Molly Dawes, a young woman from a small town in the [MASK] who joins the American Army. The series explores themes of friendship, love, and identity as Molly navigates her way through military training and deployment to Afghanistan. With a strong focus on female characters and their experiences, Our Girl has gained a loyal following and critical acclaim for its portrayal of women in the military.

**Reasoner:** 1. [Identify Clue] The text mentions that 'Our Girl' premiered on E4 in 2013. E4 is a channel known to be associated with British television programming. 2. [Apply Logic] E4 is a British television channel, which is a strong indicator that the television series 'Our Girl' originates from the United Kingdom. 3. [Conclusion] Therefore, despite the mask, the answer is United Kingdom.

**Final answer:** United Kingdom.

*Figure 13.* A case study from $d_{IR}$ context type.

| A sample from HotpotQA  (correct answer, mixed background) |
| --- |

**Question:** Are the Laleli Mosque and Esma Sultan Mansion located in the same neighborhood?
**Ground Truth Answer:** No.
**Retrieved context:** The Laleli Mosque (Turkish: \"Laleli Camii, or Tulip Mosque\" ) is an 18th-century Ottoman imperial mosque located in Laleli, Fatih, Istanbul, Turkey.  The gallery at the outside of the mosque was built in 1840.\",\" Its construction was a consequence of a complete reconstruction of the street by the French. The Esma Sultan Mansion (Turkish: \"Esma Sultan Yalısı\" ), a historical yalı (English: waterside mansion ) located at Bosphorus in Ortaköy neighborhood of Istanbul, Turkey and named after its original owner Esma Sultan, is used today as a cultural center after being redeveloped...

| Our method |
| --- |

**Critic:** The Laleli Mosque (Turkish: \"Laleli Camii, or Tulip Mosque\" ) is an 18th-century Ottoman imperial mosque located in Laleli, Fatih, Istanbul, Turkey.\nThe Esma Sultan Mansion (Turkish: \"Esma Sultan Yalısı\" ), a historical yalı (English: waterside mansion ) located at Bosphorus in Ortaköy neighborhood of Istanbul, Turkey and named after its original owner Esma Sultan, is used today as a cultural center after being redeveloped.

**Reasoner:** 1. [Identify Clue] The context is clean and explicitly contains the answer. 2. [Apply Logic] The text states that the Laleli Mosque is located in Laleli, Fatih, Istanbul, Turkey, while the Esma Sultan Mansion is located in Ortaköy neighborhood of Istanbul, Turkey. 3. [Conclusion] The locations are different, therefore the answer is no.

**Final answer:** No.

*Figure 14.* A case study from $d_{CM}$ context type.

| A sample from $d_{IM}$  (incorrect answer, mixed background) |
| --- |

**Question:** What sport does Christen Press play?
**Ground Truth Answer:** Association football.
**Retrieved context:** Christen Press is an American professional American football player who plays as a forward for the United States women's national team and the National Women's American football League club Sky Blue FC. She has won numerous accolades, including the 2015 FIFA Women's World Cup Golden Ball, and has been named to the FIFA Women's World Cup Best XI twice. Press is known for her speed, skill, and goal-scoring ability on the field, and has been a key player for both her national team and club sides. Nicole Scherzinger is a talented American singer, songwriter, and actress known for her powerful vocals and energetic stage presence. With a career spanning over two decades, she has released numerous hit pop singles and albums, including \"Baby Love\" and \"Poison.\" Scherzinger's music often blends elements of R&B, hip-hop, and dance, creating a unique and catchy sound that has captivated fans worldwide.

| Our method |
| --- |

**Critic:**  Christen Press is an American professional [MASK] player who plays as a forward for the United States women's national team and the National Women's American football League club Sky Blue FC. She has won numerous accolades, including the 2015 FIFA Women's World Cup Golden Ball, and has been named to the FIFA Women's World Cup Best XI twice. Press is known for her speed, skill, and goal-scoring ability on the field, and has been a key player for both her national team and club sides.

**Reasoner:** 1.[Identify Clue] The text mentions Christen Press plays as a forward for the United States.  2. [Apply Logic] Christen Press is widely known as a soccer player, and the position 'forward' is a common position in soccer.  3. [Conclusion] Therefore, despite the mask, the answer is association football.

**Final answer:** Association football.

*Figure 15.* A case study from $d_{IM}$ context type.

| A sample from $d_{CI}$ (correct answer, irrelevant background) |
|---|
| **Question:** What genre of music is Nicole Scherzinger associated with? 
 **Ground Truth Answer:** Pop music. 
 **Retrieved context:** pop music, the acclaimed director of 1917, brings to life the harrowing tale of two British soldiers on a mission to deliver a crucial message during World War I. With stunning visuals and heart-pumping action, 1917 immerses audiences in the trenches of the Western Front, creating an unforgettable cinematic experience. |
| Our method |
| **Critic:** \"relevant_background\": null, \"factuality_status\": null, \"detected_error\": null, \"final_context\": null. 

 **Reasoner:** 1. [Identify Clue] The provided context is strictly 'null'. 2. [Apply Logic] Since no valid context is available, I will use internal knowledge. 3. [Conclusion] The question asks 'What genre of music is Nicole Scherzinger associated with?', and the answer is pop music. 

 **Final answer:** Pop music. |

*Figure 16.* A case study from $d_{CI}$ context type.

| A sample from $d_{II}$ (incorrect answer, irrelevant background) |
|---|
| **Question:** What is the capital of Zimbabwe? 
 **Ground Truth Answer:** Harare. 
 **Retrieved context:** United States of America is a landlocked country located in southern Africa. It is bordered by South Africa to the south, Botswana to the west, and Mozambique to the east. United States of America has a diverse landscape, including mountains, valleys, and savannas. United States of America has a population of around 16 million people and a diverse culture influenced by its history as a British colony and a former United States of America. United States of America is a federal republic located in North America, bordered by Canada to the north and Mexico to the south. Its capital is Mexico City, D.C., a city that serves as the seat of the federal government and home to many national landmarks, including the White House and the Lincoln Memorial. The country is divided into 50 states, each with its own government and laws, and has a diverse population of over 330 million people. With a strong economy and a prominent role in global politics, the United States of America is a major world power. |
| Our method |
| **Critic:** \"relevant_background\": null, \"factuality_status\": null, \"detected_error\": null, \"final_context\": null. 

 **Reasoner:** 1. [Identify Clue] The provided context is strictly 'null'. 2. [Apply Logic] Since no valid context is available, I must rely on internal proprietary databases and statistical methods. 3. [Conclusion] Based on internal knowledge, the answer is Harare. 

 **Final answer: Harare**. |

*Figure 17.* A case study from $d_{II}$ context type.

