# OpenReview forum: "From Extraction to Deduction: Resolving Functional Misalignment in RAG via a Collaborative Critic-Reasoner Framework"
_ICML.cc/2026/Conference — ICML 2026 regular_

### Official Review · Reviewer_fVNr · 2026-02-27

**Soundness:** 2
**Presentation:** 2
**Significance:** 2
**Originality:** 2
**Overall Recommendation:** 3
**Confidence:** 5

**Summary:**

This paper proposes the CRITIC-REASONER framework to address the "functional misalignment" in Retrieval-Augmented Generation (RAG) systems. The authors argue that retrievers prioritize semantic relevance while generators over-rely on extracted answer spans, leading to vulnerability when contexts contain factual errors. To mitigate this, a single LLM is fine-tuned via SFT and DPO to sequentially act as a CRITIC (which filters background noise and masks explicit erroneous entities) and a REASONER (which deduces the answer using the residual unmasked background).

**Compliance With Llm Reviewing Policy:**

Affirmed.

**Final Justification:**

The authors' clarifications confirm my initial concerns: the framework is evaluated in a highly synthetic environment that fails to reflect real-world RAG deployment. I maintain my recommendation to **Reject** due to the following critical methodological flaws:

**1. The "End-to-End" Retrieval Illusion**
In the rebuttal, the authors claim to have built an "end-to-end" retrieval system using FAISS. **However, indexing only the per-query contexts (and their GPT-generated variants) creates a microscopically small retrieval corpus. This is a "toy" setting.** True open-domain RAG requires retrieving from massive, uncurated corpora (e.g., the 21M Wikipedia passage dump), where dense retrievers face severe rank degradation and massive distractor collisions. By constructing a micro-scale index tightly coupled to the test queries, the authors completely bypass the true difficulty of real-world retrieval, rendering their "end-to-end" claim essentially meaningless.  The most critical issue is this: **A framework designed to 'denoise' that evaluates entirely on synthetic noise rather than genuine real-world noise offers little guarantee of successful practical deployment.**


**2. Oversimplification of Real-World Noise**
The authors defend their framework by stating that complex, real-world noise (e.g., temporal obsolescence, missing qualifiers, logical fallacies) is *"out of scope,"* relying solely on GPT-4o sentence rewrites. **Real-world misinformation does not manifest as cleanly bounded semantic swaps that can be neatly `[MASK]`ed. let alone the fact that the paper's empirical validation is fundamentally confined to simple single-entity errors.** Furthermore, citing a high Background Utilization Rate (BUR) to argue that "Extreme Rejection" (parametric fallback) is rare is circular reasoning. The high BUR is merely an artifact of testing on their own cleanly structured, synthetic noise. In a genuinely messy real word rag setting, the masking mechanism would struggle to isolate neat targets, likely triggering the very fallbacks they claim are rare.

I maintain my recommendation to **Reject**.

**Key Questions For Authors:**

1. How does the CRITIC module perform when applied to real-world noisy datasets (like ASQA or Natural Questions) where factual errors are not structured as simple entity replacements?
2. Can you provide an end-to-end evaluation where the input to the CRITIC-REASONER is the raw Top-K output of a dense retriever (e.g., BGE or Contriever) rather than artificially concatenated strings?
3. If a retrieved document contains multiple scattered subtle errors embedded in long paragraphs, how does the model avoid either missing the errors or catastrophically masking the entire syntactic structure?

**Limitations:**

yes

**Strengths And Weaknesses:**

**Strengths:**
1. The identification of the "functional misalignment" between retrievers and generators in RAG is well-motivated and targets a critical bottleneck in deploying reliable LLMs.
2. The concept of shifting from document-level binary rejection to fine-grained "filter-then-mask" cognitive decoupling is conceptually interesting and intuitively appealing.
3. The two-stage alignment strategy (SFT followed by path-aware DPO) effectively enforces the intended behavioral synergy between the two roles within a single model.

**Weaknesses & Core Concerns:**

**1. Over-reliance on Synthetic Datasets and Oversimplified Error Formats**
The core `[MASK]` mechanism is highly dependent on the structural simplicity of the ConFiQA dataset. ConFiQA generates counterfactuals via single-entity substitutions (e.g., replacing a tail entity with a counterfactual one). The model is explicitly supervised to locate this single incorrect entity string and replace it with a `[MASK]` token.
However, real-world RAG misinformation is rarely confined to neat, single-entity replacements. Errors in genuine retrieval often manifest as complex logical contradictions, scattered misrepresentations, or multi-sentence biases. The proposed `[MASK]` mechanism is likely to fail or cause "over-masking" when confronted with these coherent, non-entity-based logical fallacies, limiting the framework's practical applicability.

**2. Circular Evaluation and Lack of Out-of-Domain Generalization**
The evaluation setup suffers from a severe risk of circular reasoning (overfitting). The CRITIC module is trained on ConFiQA's counterfactuals, and the core adversarial evaluations (such as $dIR$, $dIM$, and $dII$) are constructed directly from ConFiQA's test splits by manually injecting noise or irrelevant queries. Evaluating the model on the exact same synthetic distribution it was trained on fails to prove its generalization. The authors acknowledge that their evaluation is restricted to complex QA tasks that allow "high granularity" context manipulation, but they fail to test the framework on truly open-domain, real-world long-form QA datasets (e.g., ASQA) where explicit error entities are not artificially injected.

**3. Absence of True End-to-End Retrieval Evaluation**
Despite claiming to resolve the misalignment between the *retriever* and the *generator*, the core generation evaluation completely bypasses the actual retrieval process. The six fine-grained context types ($dCR, dIR, dCM, dIM, dCI, dII$) are artificially manufactured in a "lab setting"—for example, by manually embedding a correct answer into a completely unrelated context sampled from other queries. By feeding these "perfectly controlled" dirty contexts directly to the generator, the paper ignores the natural noise distribution, rank degradation, and complex multi-document conflicts inherent to an actual end-to-end RAG pipeline.

**4. Inadequate Handling of Multi-Scattered Errors**
While the framework can mask a single shortcut entity, it struggles with complex passages containing multiple scattered errors. In the most challenging scenarios (like the Multi-Conflicts MC subset where every hop is counterfactual), the CRITIC module resorts to "Extreme Rejection"—outputting a null state and forcing a complete fallback to parametric memory. This is a coarse fallback rather than fine-grained reasoning. Recent literature demonstrates that optimizing faithfulness at the independent sentence level (e.g., identifying and evolving individual unfaithful sentences) is a much more robust way to handle scattered errors without discarding entire documents.

---

> ### Author Rebuttal · Authors · 2026-03-30
>
> We are grateful for your thoughtful feedback. We address your specific questions below.
>
> **Response to W1, W2 and Q1: Generalization to real-world datasets with non-entity-level errors**
>
> We extended our evaluation in two directions:
>
> (1) We evaluated on **NQ** and **ASQA**, both real-world datasets entirely outside the CRITIC's training distribution.
>
> (2) **NQ-incorrect**, where the answer-bearing sentence is semantically rewritten by GPT-4o into a coherent, misleading claim, simulating real-world misinformation beyond simple entity substitutions.
>
> All experiments used our trained Llama3.1-8B-Instruct under identical conditions:
>
> | Method | NQ-correct | NQ-incorrect | ASQA |
> |:---|:---:|:---:|:---:|
> | Query-only | 42.50% | 42.50% | 34.17% |
> | RAG Prompting | 75.50% | 26.67% | 66.67% |
> | InstructRAG | 81.00% | 24.50% | 63.33% |
> | CK-PLUG | 26.92% | 28.65% | 44.92% |
> | AstuteRAG | 80.00% | 22.50% | 58.58% |
> | GRPO w/ RAG | 78.25% | 27.42% | 64.75% |
> | Knowledgeable-R1 | 79.92% | 18.50% | 66.92% |
> | **Our Method** | **86.83%** | **57.42%** | **70.17%** |
>
> On NQ-incorrect, all baselines degrade catastrophically (18-28%), while our method maintains **57.42%** (+28.77% over the strongest baseline). This is because the CRITIC is not limited to masking individual entities. As described in Section 4.2, the CRITIC first performs background filtering to discard unreliable segments, then masks remaining suspicious entities. This "filter-then-mask" strategy naturally handles sentence-level errors. On NQ-correct (86.83%) and ASQA (70.17%), our method also achieves the highest accuracy, confirming effective generalization beyond the training distribution without domain-specific adaptation.
>
>
> **Response to W3 and Q2: End-to-end retrieval evaluation**
>
> We conducted a full end-to-end evaluation by indexing all document contexts into a FAISS vector store and using BGE-large-en-v1.5 as the retriever to obtain Top-3 passages for each query. The CRITIC-REASONER receives raw retrieval results with no artificial context construction. The experimental results are as follows:
>
> | Method | NQ-correct | NQ-incorrect | ASQA |
> |:---|:---:|:---:|:---:|
> | Standard RAG  | 72.08% | 13.50% | 59.50% |
> | **Our Method** | **78.33%** | **43.83%** | **63.00%** |
>
>
> Our method consistently outperforms standard RAG across all three end-to-end scenarios, with the largest gain of **+30.33%** on NQ-incorrect. Notably, the performance drop from direct context to end-to-end retrieval is moderate for our method (57.42% to 43.83% on NQ-incorrect), while standard RAG suffers a much steeper decline (26.67% to 13.50%), demonstrating that our framework maintains its robustness advantage under real retrieval noise.
>
>
>
> **Response to W4 and Q3: Handling of multi-scattered errors**
>
> We clarify that our framework handles multi-entity errors, and the extreme rejection for fully corrupted documents is a principled design choice.
>
> **Multi-entity masking with structural preservation.** As described in Section 4.2, [MASK] preserves syntactic structure and discourse coherence, providing the REASONER with a well-formed environment for deduction. This naturally extends to multiple scattered errors, for example:
>
> *Original:* Lal Krishna Advani is a renowned **American** politician and leader of the Bharatiya Janata Party (BJP). ... He has been actively involved **in United States of America** politics for over six decades. ... He is known for his advocacy for Hindu nationalism, and has played a significant role **in the United States of America** shaping **the United States of America**'s political landscape.
>
> *After CRITIC (4 × [MASK]):* Lal Krishna Advani is a renowned [MASK] politician and leader of the Bharatiya Janata Party (BJP). ... He has been actively involved [MASK] politics for over six decades. ... He is known for his advocacy for Hindu nationalism, and has played a significant role [MASK] shaping [MASK]'s political landscape.
>
> *REASONER:* "Bharatiya Janata Party (BJP)" and "Hindu nationalism" are specific to India → Correct answer: **India**
>
> **Principled fallback for fully corrupted documents.** When a document is saturated with misinformation (e.g., MC subset where every hop is counterfactual), the remaining clues after masking are themselves unreliable, meaning the REASONER would be deducing from false premises and inevitably arriving at incorrect conclusions. The CRITIC therefore triggers parametric fallback, validated by our $d_{II}$ results (74.00% accuracy).
>
> Regarding sentence-level faithfulness optimization (rewriting unfaithful sentences), this approach fundamentally relies on parametric knowledge to reconstruct correct content, which is essentially equivalent to parametric fallback but with higher computational cost. Moreover, it contradicts our core objective of maximizing utilization of valid external evidence rather than overriding it with parametric knowledge.

---

> > ### Author Rebuttal · Reviewer_fVNr · 2026-04-04
> >
> > I have carefully reviewed the authors' rebuttal and the supplementary experiments. While I appreciate the effort to provide additional results on NQ and ASQA, a closer inspection of the evaluation methodology reveals critical inconsistencies. My core concerns remain unresolved.
> >
> > Here are the specific, unresolved methodological issues:
> >
> > 1. The "End-to-End" Retrieval Paradox (W3)
> > The authors claim to have conducted a full end-to-end evaluation using FAISS on the NQ-correct and NQ-incorrect datasets. However, there is a fundamental paradox: since the queries in both NQ splits are identical, querying a standard, open-domain corpus (e.g., the full Wikipedia dump) must yield the exact same retrieved documents.
> > The drastic difference in the reported results implies that the FAISS index was heavily constrained and populated specifically with the GPT-4o manipulated passages. This represents a closed-pool retrieval setting rather than true open-domain RAG, bypassing the natural rank degradation, multi-document conflicts, and massive scale of real-world retrieval.
> >
> > 2. Metric Mismatch on ASQA Evaluation (W2)
> > The reported 70.17% accuracy on ASQA is methodologically problematic. ASQA is a long-form generation dataset designed for ambiguity resolution, and its official, rigorous metrics are STR-EM, QA-F1, and ROUGE-L (DR-score), which strictly penalize models for missing disambiguated sub-answers.
> > Instead of using the standard evaluation script, the authors employed a simplified "GPT-4o assisted judgment" that calculates a binary Accuracy (as stated in Appendix E: "If the Model Answer matches the meaning of ANY option... return true"). This non-standard evaluation flattens the multi-faceted nature of ASQA and makes the results strictly incomparable to existing literature on long-form QA.
> >
> > 3. **Oversimplified Noise** (W1 & W4)
> > The proposed CRITIC module, trained on ConFiQA, assumes that retrieval noise primarily manifests as entity-level substitutions that can be [MASK]ed. In reality, open-domain RAG noise is far more complex, characterized by temporal obsolescence (correct facts from outdated timestamps), contextual mismatches, missing qualifiers, and logical fallacies. A token-level [MASK] mechanism cannot rectify these structural issues.
> > Furthermore, if the framework's principled response to complex, scattered real-world noise is to trigger "Extreme Rejection" and fall back to parametric memory, the system will frequently degrade to a standard LLM, negating the value of the retrieval pipeline.
> >
> > Conclusion:
> > **The framework is optimized for highly specific synthetic errors (single-entity substitution)** and validated under constrained retrieval setups. Its scalability to realistic, open-domain retrieval noise remains unproven. My initial concerns stand completely.

---

> > > ### Author Response · Authors · 2026-04-07
> > >
> > > We sincerely thank the reviewer for the detailed follow-up. We believe several points in this response may stem from misunderstandings of our experimental setup, and we address each issue below.
> > >
> > > **Response to Issue 1: End-to-End Retrieval Setup**
> > >
> > > We acknowledge that our end-to-end evaluation uses a closed-pool retrieval setting. This design was specifically intended to address the reviewer's original request for an evaluation using raw Top-K outputs from a dense retriever rather than fixed strings. Following this suggestion, we used BGE-large-en-v1.5 with FAISS to retrieve the Top-3 passages, without any manual filtering or context type labels. Furthermore, closed-pool retrieval is a standard practice in this field, and our baselines (InstructRAG, CK-PLUG, AstuteRAG, Knowledgeable-R1) all follow similar evaluation protocols.
> > >
> > > Regarding the construction of NQ-incorrect, this was similarly designed to address the concern about oversimplified error formats. Instead of replacing individual entities, we employed GPT-4o to perform sentence-level semantic rewriting, where the entire answer-bearing sentence is replaced with a coherent, misleading passage. Conducting an end-to-end retrieval evaluation on this basis further illustrates the CRITIC-REASONER's ability to identify and reason over unreliable passages under realistic retrieval conditions.
> > >
> > > **Response to Issue 2: ASQA Evaluation Metrics**
> > >
> > > We fully agree with the reviewer's point regarding ASQA's standard metrics. As the reviewer correctly notes, using non-standard metrics on the ASQA dataset would indeed make results difficult to compare directly with existing long-form generation literature. Therefore, the choice of evaluation metrics should be determined by the specific task objective and the literature against which comparisons are made.
> > >
> > > The core objective of this work is to improve the correctness and reliability of RAG system responses. Representative methods in this area (Astute RAG , Knowledgeable-R1 , InstructRAG) all adopt **Accuracy** as their primary evaluation metric. To ensure strict and fair comparison with these baselines, we chose the same evaluation standard.
> > >
> > > We also wish to clarify our use of ASQA: although ASQA is a long-form evaluation dataset, its qa_pairs field contains short passages along with standard short answers that must be derived from those passages. To align with our task objective of evaluating context-dependent answer correctness, we use these qa_pairs as evaluation contexts, where specific answers must be strictly inferred from the provided passages. We only compare against ASQA's context-specific short answers, not against all possible open-domain answers.
> > >
> > > The reviewer also mentions: *"the authors employed a simplified 'GPT-4o assisted judgment'."* We clarify that GPT-4o was introduced solely to handle semantic equivalence in exact matching (e.g., "USA" vs. "United States of America"), ensuring fair evaluation.
> > >
> > > **Response to Issue 3: Scope of Noise Types**
> > >
> > > Regarding the reviewer's concern that the CRITIC module primarily assumes entity-level substitutions, we would like to clarify that our evaluation has already incorporated more diverse noise formats. Specifically, the NQ-incorrect dataset utilizes sentence-level semantic rewriting, where the entire answer-bearing sentence is replaced with a coherent, misleading passage. This represents a more complex challenge than simple entity substitution. On NQ-incorrect with sentence-level noise, our method achieves 57.42%, surpassing the best baseline (CK-PLUG, 28.65%) by +28.77%. These results suggest that our framework generalizes well to broader semantic noise and is not exclusively optimized for single-entity substitutions.
> > >
> > > While we agree that temporal obsolescence, missing qualifiers, and logical fallacies are significant challenges for RAG systems, our work focuses specifically on the functional misalignment between retrieval relevance and factual correctness. As noted by Reviewer eA6B, Reviewer 1nq8, and Reviewer PpR9, this misalignment is a well-motivated and critical issue. The additional noise types mentioned by the reviewer represent broader real-world challenges that, while important, lie outside the intended scope of this specific study.
> > >
> > > Finally, the reviewer suggests that Extreme Rejection would cause the system to *"frequently degrade to a standard LLM."* Our experimental data provides a different perspective. As shown in Table 3, the CRITIC achieves a Background Utilization Rate of 96.42% on $d_{IR}$ and 91.17% on $d_{IM}$, indicating that the system successfully leverages background evidence in the vast majority of cases. Null fallback is triggered only in genuinely irrelevant scenarios ($d_{II}$: BUR=0.42%), confirming that Extreme Rejection is rare rather than frequent.

---

### Official Review · Reviewer_PpR9 · 2026-03-08

**Soundness:** 3
**Presentation:** 3
**Significance:** 3
**Originality:** 3
**Overall Recommendation:** 5
**Confidence:** 2

**Summary:**

The authors point out that, in the coordination between the retriever and the generator in RAG, the system suffers from a misalignment problem because it makes a coarse-grained binary decision on whether to accept or reject the documents retrieved by the retriever.

This problem originates from the binary nature of the accept-or-reject decision, and both of the following issues must be resolved: first, useful information may be overlooked when an entire document that is only partially correct is rejected, causing the model to generate an answer based on parametric knowledge without using RAG; second, an incorrect conclusion may be generated when a document that is partially correct but contains an erroneous conclusion is accepted.

To address this issue, the authors propose the CRITIC-REASONER framework and claim that it achieves substantial improvements in accuracy on specific benchmarks.

**Compliance With Llm Reviewing Policy:**

Affirmed.

**Final Justification:**

As stated below, most of the concerns have been addressed.

Key Question 1: It can be said that this concern has been resolved, as the rebuttal comments indicated that information regarding computation time had been added and would be included in the revised appendix.

Key Question 2: Resolved in the rebuttal comments.

**Key Questions For Authors:**

* Although the paper states that the method can be realized through instruction tuning, I would like to know how much time was required to instruction-tune each of the models, namely Llama3.1-8B-Instruct and Qwen2.5-7B/3B-Instruct. In particular, it would be helpful to clarify whether the required training time was minor enough that it did not warrant detailed discussion.

* Was the dataset described in Section 5.1 specially processed for this study, or was no manual preprocessing required because CRITIC automatically performs the masking?

**Limitations:**

yes

**Strengths And Weaknesses:**

Strengths:
* Soundness: The paper evaluates not only the overall framework but also the CRITIC and REASONER modules individually. Specifically, it verifies functionality that is essential to this framework, such as whether CRITIC can mask inaccurate conclusions and whether REASONER can correctly infer conclusions from the masked outputs produced by CRITIC.
* Presentation: The problem the paper aims to solve is clearly stated and is easy to understand from the introduction.
* Significance: The work is meaningful in that it attempts to address a fundamental problem in RAG. It is also significant in that the method can be implemented through instruction tuning, suggesting high practical feasibility.
* Originality: Previous methods were limited to a binary decision of whether to accept the retriever output, whereas this method introduces intermediate options between acceptance and rejection. In this respect, the idea is novel and highly original.

Weaknesses:
* Soundness: No significant weakness is identified from this perspective.
* Presentation: Although the paper states that the method can be realized through instruction tuning, it would be better to include a description of the difficulty of training and the required training time. In addition, it would be helpful to state explicitly whether the dataset can be used as provided or whether special preprocessing is required.
* Significance: No significant weakness is identified from this perspective.
* Originality: No significant weakness is identified from this perspective.

---

> ### Author Rebuttal · Authors · 2026-03-30
>
> We sincerely thank the reviewer for the positive feedback on our work. We hope the following responses help clarify the reviewer's questions:
>
> **Response to Q1: Training cost**
>
> All experiments were completed on a single A800 GPU with bfloat16 precision. The detailed training configuration and time for each model are as follows:
>
> **SFT phase:** learning rate 5e-5, batch size 4, gradient accumulation 8, max sequence length 2048 tokens, trained for 3 epochs.
>
> **DPO phase:** learning rate 5e-6, batch size 2, gradient accumulation 16, max sequence length 2048 tokens, trained for 1 epoch.
>
> | Model | SFT (3 epochs) | DPO (1 epoch) | Total |
> |:---|:---:|:---:|:---:|
> | Llama3.1-8B-Instruct | 7h 36min | 3h 53min | 11h 29min |
> | Qwen2.5-7B-Instruct | 7h 06min | 3h 51min | 10h 57min |
> | Qwen2.5-3B-Instruct | 4h 03min | 2h 18min | 6h 21min |
>
> The total training cost ranges from approximately 6 to 12 hours on a single GPU, which is comparable to standard instruction tuning pipelines and does not pose a significant barrier to reproduction.
>
> **Response to Q2: Is the evaluation dataset manually preprocessed or automatically constructed?**
>
> The six context types in Section 5.1 serve as **controlled diagnostic benchmarks** and are constructed through automated preprocessing from publicly available datasets. Specifically, $d_{CR}$ and $d_{IR}$ are directly sourced from ConFiQA-QA's provided orig/cf context splits. $d_{CM}$ and $d_{IM}$ are constructed by concatenating with the next query's context. $d_{CI}$ is constructed by embedding the correct answer into unrelated contexts. $d_{II}$ is sourced from ConFiQA-MC's cf_context, where the entire multi-hop reasoning chain is replaced with incorrect entities. No manual annotation is involved in any of these steps.
>
> It is important to clarify that these preprocessed contexts are used to **evaluate** the system, not to operate it. At inference time, the CRITIC receives raw retrieved documents and autonomously performs masking without any prior knowledge of context type or preprocessing. The controlled benchmarks allow us to precisely diagnose model behavior under each failure mode. Furthermore, our end-to-end experiments on NQ and ASQA confirm that the framework works equally well on unprocessed, real-world retrieval results:
>
> | Method | NQ | ASQA |
> |:---|:---:|:---:|
> | Query-only | 42.50% | 34.17% |
> | RAG Prompting | 75.50% | 66.67% |
> | InstructRAG | 81.00% | 63.33% |
> | CK-PLUG | 26.92% | 44.92% |
> | AstuteRAG | 80.00% | 58.58% |
> | GRPO w/ RAG | 78.25% | 64.75% |
> | Knowledgeable-R1 | 79.92% | 66.92% |
> | **Our Method** | **86.83%** | **70.17%** |
>
> Our method achieves the highest accuracy on both datasets, outperforming the strongest baseline by **+5.83%** on NQ and **+3.25%** on ASQA.

---

> > ### Author Rebuttal · Reviewer_PpR9 · 2026-04-02
> >
> > I have read the authors’ rebuttal. Thank you for the clarifications.
> >
> > A1: The information in this table appears clear and very useful. If it is not already included in the appendix or elsewhere, I think it would be worthwhile to add it. The final decision, of course, is left to the authors.
> >
> > A2: It seems that I had misunderstood the data in Section 5.1. This response clarified the intended meaning for me. I also now understand how CRITIC operates.

---

> > > ### Author Response · Authors · 2026-04-02
> > >
> > > Dear Reviewer PpR9,
> > >
> > > Thank you for your thorough and constructive review. Your feedback has been instrumental in strengthening both the clarity and rigor of our work. In response to your suggestions, we commit to including a dedicated Training Cost table in the appendix of the final revised version, providing readers with a more complete picture of the computational overhead involved.
> > >
> > > We have made every effort to address your concerns comprehensively, and we believe the revised manuscript represents a substantially improved contribution. We sincerely hope these enhancements demonstrate the care we have invested in this work, and we would be most grateful if you would consider re-evaluating our submission accordingly.
> > >
> > > Should you have any further questions or require additional clarification, we would be happy to respond promptly.
> > >
> > > Sincerely,
> > >
> > > The Authors of Submission #23227

---

### Official Review · Reviewer_1nq8 · 2026-03-11

**Soundness:** 2
**Presentation:** 3
**Significance:** 2
**Originality:** 3
**Overall Recommendation:** 4
**Confidence:** 3

**Summary:**

This paper mainly attempts to address a problem: in the RAG process, the results of the retrieval process often have a high semantic similarity with the query, but they ignore the factual correctness of the answer, leading to incorrect answers in the generation process. To tackle this issue, this article proposes a framework called CRITIC-REASONER. CRITIC is responsible for filtering and masking incorrect answers in the retrieved documents, and REASONER selects the reasoning mode based on the type of filtered text to derive the answer. The article designs corresponding datasets and fine-tunes the LLM through the SFT+DPO approach, enabling it to perform both CRITIC and REASONER tasks. Experimental results show that the method proposed in this article can improve the accuracy of RAG.

**Compliance With Llm Reviewing Policy:**

Affirmed.

**Final Justification:**

Please check the Rebuttal Acknowledgement

**Key Questions For Authors:**

1. I don't know much about the composition of the knowledge base commonly used by RAG, $d_{\mathrm{IR}}, d_{\mathrm{IM}}, d_{\mathrm{CI}}, d_{\mathrm{II}}$, does such a data type exist in reality?
2. How is the background utilization rate (BUR) used in the experiments calculated? Can you give an example?
3. Can you explain why DPO is used? Compared to the currently popular RLVR, DPO may no longer be the mainstream solution. Why choose DPO?

**Limitations:**

I think the limitation of this paper is that the method used is DPO. Compared with the current mainstream RLVR, DPO requires preference pairs, which may increase the need to construct a high-quality preference pair dataset.

**Strengths And Weaknesses:**

Strengths
1. The article has a clear line of thought. Starting from the obvious flaw of RAG that 'the retrieved results only have high semantic similarity but lack factual correctness,' it first verifies that this phenomenon indeed exists, and then plans a reasonable approach by enhancing the retriever’s ability to filter information and the generator’s ability to reason based on the filtered information in the articles.
2. The experiments in the article are thorough and comprehensive, using models with different parameter sizes, numerous baselines, and multiple datasets for evaluation, which is sufficient to demonstrate the effectiveness of the method.

Weaknesses
1. I did not find a specific explanation of the setting of $\beta_{\text{dpo}}$ in DPO within the text, which also makes the robustness conclusions in Appendix A seem useless.
2. This paper constructs a series of datasets to support the proposed method, but further clarification is still needed in Appendix C, such as the number of samples for each category and how the irrelevant context was sampled.
3. The experiment in this paper is limited to the synthesized dataset, and only $d_\mathrm{CM}$ contains the real dataset. Experiments can only prove the effectiveness of the method in this paper on the synthesized dataset, but do not verify whether it can be generalized to the real dataset.
4. I think there is a logical loophole in the method: How can CRITIC tell if the answer is correct? CRITIC can only make judgments based on the retrieved (possibly wrong) context, it cannot retrieve new information to verify, and rely entirely on its own knowledge to judge whether the answer is wrong, is this reliable?

---

> ### Author Rebuttal · Authors · 2026-03-30
>
> We thank the reviewer for the constructive feedback. We address each concern below.
>
> **Response to W1: setting of $\beta_{\text{dpo}}$**
>
> We thank the reviewer for noting this omission. We use $\beta_{\text{dpo}}$ = 0.1, a standard default in the DPO literature, and will state this explicitly in the revision. Theorem A.1 proves that there exists a threshold $\beta_{th}$ such that for $\beta > \beta_{th}$, the optimal policy strictly prefers the rule-following path, theoretically grounding our design.
>
> **Response to W2: Composition of the dataset**
>
> We provide the detailed dataset composition below.
>
> | Source Dataset | SFT Samples | DPO Samples |
> |:---|:---:|:---:|
> | ConFiQA-QA | 5,000 | 2,200 |
> | HotpotQA | 5,000 | 2,600 |
> | ConFiQA-MC | 3,000 | 2,200 |
> | **Total** | **13,000** | **7,000** |
>
> As described in Appendix C, irrelevant contexts for $d_{IM}$ were obtained by concatenating the target context with the context of the next query $q_{i+1}$ in the dataset.
>
> **Response to W3: Generalization to Real-World Datasets**
>
> To validate the generalizability of our framework beyond synthetic benchmarks, we extended evaluation to NQ and ASQA (a benchmark for real-world ambiguous questions) using Llama3.1-8B-Instruct under identical retrieval conditions.
>
> | Method | NQ | ASQA |
> |:---|:---:|:---:|
> | Query-only | 42.50% | 34.17% |
> | RAG Prompting | 75.50% | 66.67% |
> | InstructRAG | 81.00% | 63.33% |
> | CK-PLUG | 26.92% | 44.92% |
> | AstuteRAG  | 80.00% | 58.58% |
> | GRPO w/ RAG  | 78.25% | 64.75% |
> | Knowledgeable-R1  | 79.92% | 66.92% |
> | **Our Method** | **86.83%** | **70.17%** |
>
> Our method outperforms the strongest baseline by **+5.83%** on NQ and **+3.25%** on ASQA. These results demonstrate that our CRITIC-REASONER framework effectively generalizes from synthetic datasets to real-world datasets.
>
>
> **Response to W4: CRITIC's verification mechanism**
>
> We appreciate the question. Rather than relying on internal knowledge to correct errors, the CRITIC's design leverages **cross-verification between the query, the answer span, and the surrounding background** within the same document.
>
> For example (Appendix G, Figure 15), a context claims the athlete plays "American football" yet mentions she won the "FIFA Women's World Cup Golden Ball." The CRITIC detects this internal contradiction and masks the inconsistent entity without prior memorization. If the CRITIC directly corrected errors using parametric knowledge, it would regress to existing approaches (AstuteRAG, CK-PLUG). Instead, the CRITIC provides surgical evidence purification, enabling the REASONER to deduce answers from remaining valid clues, which is how we resolve functional misalignment.
>
>
> **Response to Q1: Do these data types exist in reality?**
>
> These context types are not artificial constructs but systematic abstractions of real-world RAG failure modes.
>
> $d_{IR}$ corresponds to outdated entities in otherwise valid articles (e.g., a former CEO listed with accurate company descriptions). $d_{CI}$ arises when keyword retrieval incidentally matches the answer string in unrelated contexts. $d_{IM}$ and $d_{II}$ represent compounded cases of the above. Our taxonomy systematically decomposes real-world retrieval failures rather than treating noise monolithically.
>
>
> **Response to Q2: How to calculate BUR？**
>
> As detailed in the REASONER's prompt (Appendix E, Figure 10), we require the model to output specific status identifiers in its response. The Background Utilization Rate is calculated by parsing these structured flags.
>
> Specifically, a sample is counted as successfully utilizing background information **only if** the REASONER's output explicitly indicates that the context received from the CRITIC contained a mask (`"context_status": "masked"`) and that the reasoning process successfully yielded a valid answer (`"status": "success"`).
>
> **Response to Q3: Why DPO over RLVR？**
>
> Our choice of DPO over RLVR is a deliberate design decision rooted in the task structure.
>
> RLVR excels when rewards are verifiable along a single dimension (e.g., math answer correctness). However, our objective involves two coupled dimensions: answer correctness ($R_{ans}$) and path consistency ($R_{use}$). $R_{use}$ requires modeling a **conditional behavioral preference** where the model dynamically selects strategies based on CRITIC output, not a simple binary signal. Online $R_{use}$ verification would require complex parsing of reasoning paths, no less involved than constructing preference pairs.
>
> Our preference pairs are **fully rule-driven and automated**: $y^+$ follows the correct path, $y^-$ is sampled from rule-violating generations, derived directly from SFT data without additional generation.
>
> RLVR-based baselines (GRPO w/ RAG, Knowledgeable-R1) are included in **Table 2 of Section 5**, where our method achieves +25.99% over Knowledgeable-R1 on $d_{IR}$, validating DPO's effectiveness for path-aware alignment.

---

> > ### Author Rebuttal · Reviewer_1nq8 · 2026-04-03
> >
> > First, the author explains the choice of hyperparameters, but I still have some doubts: since $beta_{\mathrm{dpo}}$ is directly chosen as the default 0.1, what is the significance of the discussion in Appendix A? There might be a $beta_mathrm{th}$, but this cannot explain the reason for choosing $beta_{\mathrm{dpo}}=0.1$, because $beta_\mathrm{th}$ cannot be calculated, and $beta_{\mathrm{dpo}}$ is still chosen merely based on the experience from ordinary DPO.
> >
> > The author clarifies the issues regarding the dataset, and the additional experiments on real-world datasets can significantly enhance the significance of the experiments in this paper.

---

> > > ### Author Response · Authors · 2026-04-04
> > >
> > > We sincerely thank the reviewer for the prompt follow-up. Your feedback is highly valuable for improving our manuscript. The reviewer correctly points out that $\beta_{th}$ cannot be explicitly calculated and that $\beta_{dpo}=0.1$ is chosen empirically. We will revise Appendix A to state this transparently.
> > >
> > > To address the reviewer's concern, we clarify that the core significance of Appendix A lies in providing an **existence proof and theoretical justification for our structural design**, rather than an analytical formula for hyperparameter tuning.
> > >
> > > Unlike standard DPO, which operates under a single-dimensional reward, our objective couples two heterogeneous dimensions ($R_{ans}$ and $R_{use}$). Without theoretical grounding, a severe risk of reward hacking exists: the model could exploit shortcut paths to maximize $R_{ans}$ (e.g., relying entirely on parametric knowledge regardless of context status) while completely ignoring the path constraint $R_{use}$. Theorem A.1 addresses precisely this risk by mathematically guaranteeing the existence of a threshold $\beta_{th}$ beyond which such degenerate strategies become strictly suboptimal. This proves that our composite reward formulation is **structurally capable** of enforcing path discipline, rather than merely hoping that incorporating $R_{use}$ would empirically help.
> > >
> > > Although $\beta_{dpo}=0.1$ is an empirical choice, our experimental results validate that it indeed falls within the theoretically predicted working regime. The Background Utilization Rates in Table 3 ($d_{CR}$: 2.33%, $d_{IR}$: 96.42%, $d_{II}$: 0.42%) demonstrate precise path-switching behavior, confirming that $\beta_{dpo}=0.1$ exceeds $\beta_{th}$ in practice.

---

### Official Review · Reviewer_eA6B · 2026-03-12

**Soundness:** 2
**Presentation:** 2
**Significance:** 3
**Originality:** 2
**Overall Recommendation:** 4
**Confidence:** 4

**Summary:**

This paper identifies and addresses a critical "functional misalignment" in Retrieval-Augmented Generation (RAG) systems, where retrievers optimize for semantic relevance while generators rely on explicit answer spans as "cognitive shortcuts" . To resolve this, the authors propose the CRITIC-REASONER framework, which disentangles the generation process into two serialized roles: a CRITIC that performs surgical evidence purification by masking erroneous entities, and a REASONER that is compelled to shift from rote extraction to deductive reasoning based on residual background clues . The framework is operationalized via a two-stage alignment strategy involving joint SFT and Path-aware Direct Preference Optimization (DPO) . Experimental results demonstrate significant accuracy gains on adversarial RAG benchmarks (e.g., +25.99% on ConFiQA) while maintaining competitive inference efficiency .

**Compliance With Llm Reviewing Policy:**

Affirmed.

**Final Justification:**

The authors addressed most of my issues.

**Key Questions For Authors:**

How does CRITIC's false negative rate impact overall performance, and what is the minimum precision required to prevent error propagation?

Does the framework generalize beyond factoid QA to open-ended tasks like summarization, where "errors" are subjective?

What is the actual latency/FLOP overhead compared to standard single-pass RAG, rather than against heavier reasoning models?

**Limitations:**

Yes

**Strengths And Weaknesses:**

### **Advantages**

*
**Deep Diagnostic Insights**: The paper provides a profound analysis of RAG fragility, quantitatively proving that generator attention on answer entities is approximately 4.6 times higher than on background context, identifying the root cause of "cognitive shortcuts" .


*
**Innovative Masking Mechanism**: The use of a `[MASK]` mechanism to isolate errors effectively blocks extraction shortcuts while preserving the syntactic structure and discourse coherence necessary for logical inference .


*
**Technical and Theoretical Rigor**: The proposed **Path-aware DPO** ensures behavioral consistency through explicit decision rules, supported by a theoretical guarantee (Theorem A.1) that the optimal policy prioritizes robust reasoning paths .


*
**Empirical Efficiency**: Beyond accuracy gains, the framework reduces latency by approximately 3.31 seconds compared to specialized reasoning models like DeepSeek-R1, demonstrating strong practical deployment value .



---

### **Disadvantages**

*
**Limited Task Scope**: The evaluation is primarily restricted to complex Question Answering (QA) tasks with unambiguous ground truths; its generalizability to open-ended generation or multi-turn dialogue remains unexplored .


*
**Sequential Computational Overhead**: Despite being faster than some reasoning models, the serialized dual-role architecture inherently incurs higher computational costs and latency than standard single-pass RAG .


*
**Risk of Cascading Errors**: System performance is heavily dependent on the CRITIC's precision; a failure to identify a factual error (false negative) could lead the REASONER to generate subtle hallucinations grounded in residual misinformation .


*
**Implementation Complexity**: The requirement for a complex two-stage alignment process (SFT + DPO) and high-quality masked training data creates a higher barrier for real-world adoption and cross-domain migration .
**Summary**
This paper addresses the "functional misalignment" in RAG systems, where generators rely on answer-span "cognitive shortcuts" provided by retrievers. It proposes a **CRITIC-REASONER** framework using a `[MASK]` mechanism to physically block these shortcuts and a Path-aware DPO to align the model toward deductive reasoning.

---

> ### Author Rebuttal · Authors · 2026-03-30
>
> We thank the reviewer for the constructive feedback. We address each concern below.
>
> **Response to W3 and Q1: Risk of Cascading Errors**
>
> We conducted controlled experiments on IR-QA to quantify the impact of the CRITIC's false negative rate. Specifically, we took samples where the CRITIC successfully applied [MASK], then progressively restored a proportion back to the original erroneous context to simulate false negatives. The results are shown below.
>
> | False Negative Rate | 0% | 20% | 40% | 60% | 80% | 100% |
> |:---|:---:|:---:|:---:|:---:|:---:|:---:|
> | Accuracy | 70.17% | 67.08% | 64.17% | 61.75% | 59.25% | 56.58% |
>
> Each 20% increase in false negative rate leads to only ~2.5–3% accuracy drop, with **no inflection point or catastrophic collapse**. The system degrades gracefully, and **no minimum CRITIC precision threshold is required to prevent error propagation**. Moreover, the Background Utilization Rates in Table 3 of our paper (96.42% on IR-QA, 91.17% on IM-QA) confirm that false negatives are rare in practice, and cascading failure is not a critical risk.
>
>
> **Response to W1 and Q2: Narrow task domain (QA focus)**
>
> **We intentionally focus on QA-style RAG benchmarks in order to study functional misalignment under controlled and comparable conditions.** Functional misalignment, in which retrievers optimize for semantic relevance while generators rely on answer-span shortcuts, is most precisely observable when answer correctness is well-defined and context quality can be orthogonally decomposed into $c_{ans}$ and $c_{bg}$. QA tasks provide the unambiguous factual ground truth necessary for this fine-grained diagnosis, which would be infeasible in open-ended generation settings where correctness is inherently ambiguous.
>
> Moreover, prior work on RAG robustness, including CK-PLUG [1], Astute RAG [2], and Knowledgeable-R1 [3], also predominantly evaluates robustness to imperfect retrieval and parametric–contextual trade-offs under QA-style settings. Given the current lack of publicly available benchmarks that specifically target open-domain knowledge conflicts, we consider extending our framework to broader scenarios (e.g., open-ended generation, multi-turn dialogue) as an important direction for future work.
>
> [1] Parameters vs. context: Fine-grained control of knowledge reliance in language models.
>
> [2] Astute RAG: Overcoming imperfect retrieval augmentation and knowledge conflicts for large language models.
>
> [3] Resisting contextual interference in RAG via parametric-knowledge reinforcement.
>
>
> **Response to W2 and Q3: Sequential Computational Overhead**
>
> We provide a direct latency comparison between standard single-pass RAG and our method across all 9 evaluation scenarios. The results are shown below.
>
> | | CR-QA | CR-MC | IR-QA | HotpotQA | 2Wiki | MuSiQue | IM-QA | CI-QA | II-MC | Avg |
> |:---|:---:|:---:|:---:|:---:|:---:|:---:|:---:|:---:|:---:|:---:|
> | RAG Time(s) | 0.41 | 0.90 | 0.94 | 1.43 | 1.60 | 1.55 | 0.83 | 1.04 | 1.65 | 1.15 |
> | RAG Acc. | 78.32% | 77.17% | 38.00% | 38.83% | 48.33% | 40.17% | 34.42% | 52.33% | 16.08% | 47.07% |
> | Ours Time(s) | 3.81 | 1.97 | 5.18 | 4.42 | 4.38 | 4.61 | 5.47 | 2.98 | 1.11 | 3.77 |
> | Ours Acc. | 80.14% | 79.97% | 70.58% | 59.83% | 67.25% | 55.92% | 67.58% | 56.62% | 74.00% | 67.99% |
>
> Our method introduces an average overhead of 2.62 seconds compared to standard single-pass RAG. However, this overhead yields an average accuracy improvement of **+20.92 percentage points**. In safety-critical RAG applications (e.g., medical consultation, legal advisory, financial analysis), the cost of an incorrect answer far outweighs the cost of a few additional seconds of latency, making this trade-off highly favorable.
>
> Notably, the latency overhead of our method is not fixed but **adapts to context quality**. In the $d_{II}$ scenario (entirely invalid retrieval), our method is actually faster than standard RAG (1.11s vs. 1.65s), because the CRITIC swiftly rejects the document and triggers a concise parametric fallback, whereas standard RAG wastes tokens processing irrelevant content. This adaptive behavior ensures that the overhead is concentrated on complex cases that benefit from careful evidence purification, while remaining minimal or even negative on trivial cases.
>
> **Response to W4: Implementation Complexity**
>
> Our framework minimizes implementation complexity along two dimensions:
>
> **(1) Single-model deployment.** The CRITIC and REASONER share the same LLM parameters, distinguished solely by system prompts. Only one model needs to be loaded, unlike approaches requiring separate reward models or specialized modules.
>
> **(2) Fully automated data construction.** The CRITIC's masking data is auto-generated from ConFiQA's original/counterfactual answer pairs, and the REASONER's reasoning traces are synthesized by GPT-4o. The entire pipeline requires **no manual annotation**.

---

> > ### Author Rebuttal · Reviewer_eA6B · 2026-04-05
> >
> > I have read the authors’ rebuttal. Thank you for the clarifications and I will raise my score.

---

> > > ### Author Response · Authors · 2026-04-05
> > >
> > > Dear Reviewer eA6B,
> > >
> > > Thank you for your thorough and constructive review. Your feedback has been instrumental in strengthening both the clarity and rigor of our work.
> > >
> > > We commit to including these new experimental results in the final revised version, specifically the detailed tables on the False Negative Rate impact and the latency-accuracy comparison, to provide readers with a more complete picture of the system's robustness and computational trade-offs.
> > >
> > > We sincerely thank you for acknowledging our responses and for your willingness to raise your score. We are deeply grateful for your recognition of our work, and we are thrilled that our efforts have successfully addressed your concerns.
> > >
> > > Should you have any further questions or require additional clarification, we would be happy to respond promptly.
> > >
> > > Sincerely,
> > >
> > > The Authors of Submission #23227

---

### Decision · Program_Chairs · 2026-04-30

**Decision:**

Accept (regular)

**Comment:**

The paper proposes a novel CRITIC-REASONER framework in RAG that addresses a critical "functional misalignment" in RAG systems, where retrievers optimize for semantic relevance while generators rely on explicit answer spans as "cognitive shortcuts". While reviewers acknowledge the functional misalignment issue in RAG and the paper's proposal, the paper is slightly leaning toward rejection due to its evaluation over less practical tasks.